# Multilevel Analysis of Zero-Dose Children in Sub-Saharan Africa: A Three Delays Model Study

**DOI:** 10.3390/vaccines13090987

**Published:** 2025-09-21

**Authors:** Charles S. Wiysonge, Muhammed M. B. Uthman, Duduzile Ndwandwe, Olalekan A. Uthman

**Affiliations:** 1Cochrane South Africa, South African Medical Research Council, Francie van Zijl Drive, Parrow Valley, Cape Town 7501, South Africa; duduzile.ndwandwe@mrc.ac.za; 2Department of Epidemiology & Community Health, University of Ilorin, Ilorin 240003, Kwara State, Nigeria; uthman.mb@unilorin.edu.ng; 3Warwick Centre for Global Health Research, Division of Health Sciences, Warwick Medical School, The University of Warwick, Coventry CV4 7AL, UK; olalekan.uthman@warwick.ac.uk

**Keywords:** zero-dose children, multilevel analysis, sub-Saharan Africa, health systems

## Abstract

Background: Zero-dose children represent a critical challenge for achieving universal immunization coverage in sub-Saharan Africa. This study applies the Three Delays Model to examine multilevel factors associated with zero-dose children. Methods: We analyzed data from 30,500 children aged 12–23 months across 28 sub-Saharan African countries using demographic and health surveys (2015–2024). Zero-dose status was defined as not receiving the first dose of diphtheria–tetanus–pertussis vaccine. Multilevel logistic regression models examined individual-, community-, and country-level determinants. Results: Overall, zero-dose prevalence was 12.19% (95% confidence interval: 11.82–12.56), ranging from 0.51% in Rwanda to 40.00% in Chad. Poor maternal health-seeking behavior showed the strongest association (odds ratio (OR) 12.00, 95% credible interval: 9.78–14.55). Paternal education demonstrated clear gradients, with no formal education increasing odds 1.52-fold. Maternal empowerment factors were significant: lack of decision-making power (OR = 1.23), financial barriers (OR = 1.98), and no media access (OR = 1.32). Low community literacy and low country-level health expenditure were associated with increased zero-dose prevalence. Substantial clustering persisted at community (19.5%) and country (18.7%) levels. Conclusions: Zero-dose children concentrate among the most disadvantaged populations, with maternal health-seeking behavior as the strongest predictor. Immediate policy actions should integrate antenatal care with vaccination services, target high-parity mothers, eliminate financial barriers, and increase health expenditure to 15% of national budgets.

## 1. Introduction

Childhood immunization represents one of the most successful and cost-effective public health interventions, preventing an estimated 4–5 million deaths annually and generating substantial economic returns [1]. Despite remarkable global progress in vaccination coverage over the past decades, significant disparities persist, with the burden of unvaccinated children disproportionately concentrated in low- and middle-income countries [2]. The World Health Organization (WHO) defines “zero-dose children” as those who have not received any routine vaccines, typically measured by the absence of the first dose of diphtheria–tetanus–pertussis (DTP1) vaccine [3]. These children represent the most marginalized populations and serve as a critical indicator of health system equity and reach.

Sub-Saharan Africa bears a disproportionate burden of zero-dose children, accounting for approximately 60% of the global total despite representing only 17% of the world’s population [4]. The COVID-19 pandemic has exacerbated existing vaccination inequities, with routine immunization services experiencing significant disruptions across the region [5]. The Immunization Agenda 2030 has prioritized reaching zero-dose children as a cornerstone strategy for achieving universal immunization coverage and advancing health equity [3]. However, addressing this challenge requires a comprehensive understanding of the complex multilevel factors that create and perpetuate vaccination inequities.

Traditional approaches to understanding vaccination coverage have often focused on individual-level characteristics, such as maternal education, socioeconomic status, and healthcare access [6]. However, emerging evidence from global vaccination initiatives, including lessons from polio eradication campaigns and measles elimination efforts, demonstrates that contextual factors at community and country levels play equally critical roles in shaping vaccination outcomes [7]. Studies from successful vaccination programs, highlight the importance of addressing multilevel barriers simultaneously [8].

The Three Delays Model, originally developed to understand maternal mortality, provides a comprehensive framework for examining barriers to healthcare access and utilization [9]. This model conceptualizes three critical delays: the decision to seek care (first delay), reaching care (second delay), and receiving quality care (third delay). Recent applications of this framework to childhood vaccination have demonstrated its utility in identifying multilevel barriers and informing targeted interventions [10]. The first delay encompasses factors influencing parental decision-making about vaccination, including health knowledge, cultural beliefs, and maternal empowerment. The second delay addresses barriers to accessing vaccination services, such as geographic distance, transportation costs, and service availability. The third delay focuses on the quality of care received, including provider competence, vaccine availability, and service organization [11].

Multilevel analytical approaches have gained prominence in vaccination research, recognizing that individual behaviors are embedded within community and country contexts that shape health outcomes [12]. These methods allow for simultaneous examination of individual compositional effects and contextual area effects, providing insights into how place-based factors influence health behaviors beyond individual characteristics [13]. Recent multilevel studies have revealed substantial community- and country-level variation in vaccination coverage, highlighting the importance of contextual interventions alongside individual-focused strategies [14].

Despite growing recognition of multilevel influences on vaccination coverage, comprehensive analyses examining the simultaneous effects of individual-, community-, and country-level factors on zero-dose children in sub-Saharan Africa remain limited. Most existing studies focus on single countries or examine only individual-level determinants, limiting our understanding of the complex interplay of factors that create vaccination inequities across diverse contexts [15]. Furthermore, few studies have applied theoretical frameworks such as the Three Delays Model to systematically examine barriers to vaccination access and utilization across multiple levels.

This study addresses these knowledge gaps by conducting a comprehensive multilevel analysis of zero-dose children across sub-Saharan African countries, using the Three Delays Model as a guiding framework. The primary aim of this study was to determine the prevalence and multilevel determinants of zero-dose children across sub-Saharan African countries using the Three Delays Model framework. The secondary aims were to: (1) quantify the relative contribution of individual-, community-, and country-level factors to zero-dose status; (2) assess the extent of clustering in zero-dose patterns at community and national levels; and (3) identify modifiable factors that could inform targeted interventions to reduce zero-dose children in the region.

## 2. Methods

### 2.1. Study Design

This study employed a cross-sectional analytical design using nationally representative household survey data from the most recent demographic and health surveys (DHS) conducted across sub-Saharan Africa [16]. The cross-sectional design enabled examination of associations between individual-, community-, and country-level factors and zero-dose vaccination status at a single point in time across diverse sub-Saharan African contexts. Data were analyzed using multilevel modeling techniques to account for the hierarchical structure of individuals nested within communities and countries.

### 2.2. Setting

The study was conducted across 28 sub-Saharan African countries that met inclusion criteria for recent survey availability and data completeness. These countries represent diverse geographic regions (East, West, Central, and Southern Africa), economic development levels, health system structures, and vaccination program performance. The included countries span a range of contexts from post-conflict settings to stable democracies, from least developed countries to middle-income nations, providing broad representativeness of the sub-Saharan African region. Survey data were collected between 2015 and 2024, capturing contemporary vaccination patterns and determinants across this diverse geographical and temporal landscape.

### 2.3. Participants

The study population comprised children aged 12–23 months from households sampled in DHSs across the 28 included countries. This age group was selected as it represents the optimal window for assessing zero-dose status according to WHO recommendations, allowing sufficient time for receipt of first-dose vaccines whilst minimizing maternal recall bias. In households with multiple eligible children, only the youngest child aged 12–23 months was included to ensure statistical independence and avoid over-representation of higher-parity mothers. Children were excluded if vaccination status could not be determined from either vaccination cards or reliable maternal recall. Countries were excluded if they lacked recent surveys within the specified timeframe. The final analytical sample included 30,500 children nested within 15,637 communities across 28 countries, providing adequate statistical power for multilevel analyses and robust estimates of associations across different levels of the data hierarchy.

### 2.4. Data Sources and Measurement

Data were obtained from the DHS Program (www.dhsprogram.com, accessed on 18 August 2025), which provides free access to nationally representative household survey datasets for research purposes following user registration. The DHS employs standardized questionnaires and protocols across countries, ensuring comparability of measurements and definitions. Vaccination data were collected through a dedicated immunization module within the Children’s Questionnaire using a hierarchical approach: first from vaccination cards when available, then from maternal recall when cards were unavailable. Interviewers systematically recorded vaccination information for each recommended vaccine, including specific dates when documented or confirmation of receipt when dates were unavailable. Household and maternal characteristics were obtained from the Women’s Questionnaire, whilst community-level variables were constructed by aggregating individual responses within primary sampling units. All datasets used in this analysis are publicly available through the DHS Program’s online data portal. Country-level indicators were obtained from World Bank databases, UN Development Programme reports, and WHO health expenditure databases, matched to survey years to ensure temporal alignment.

### 2.5. Sampling Technique

The DHS employs a standardized multi-stage stratified cluster sampling methodology across all participating countries. The sampling frame is based on the most recent national population census in each country, ensuring representativeness of the target population. The multi-stage sampling process involves several sequential steps designed to achieve nationally representative samples while maintaining statistical efficiency.

### 2.6. Community and Neighborhood Definitions

We defined communities as geographical clusters based on shared primary sampling units (PSUs) within the DHS data structure. DHS sampling frames utilize the most recent national census to identify PSUs, which serve as the foundation for community-level analysis. Urban PSUs typically correspond to census enumeration blocks, while rural PSUs are generally defined by village boundaries or other recognized administrative units. This analytical unit was selected for two primary considerations. First, PSUs provide the most standardized community measure across all surveys, making them optimal for cross-regional comparative analysis [17,18]. Second, research demonstrates that DHS cluster sample sizes typically achieve adequate precision with minimal statistical bias, with cluster-based estimates showing only approximately 4% bias when using 25 women per cluster as population proxies [17,18]. Throughout this study, we use community and neighborhood terminology interchangeably to refer to these PSU-based geographical units.

### 2.7. Outcome Variable

The primary outcome variable was zero-dose status among children aged 12–23 months, defined according to WHO recommendations as children who have not received the first dose of diphtheria–tetanus–pertussis (DTP1) vaccine. The outcome variable was coded as a binary indicator, where 1 represented zero-dose status (child had not received DTP1) and 0 represented vaccinated status (child had received at least DTP1). The 12–23 month age range was selected because children in this age group should have had adequate opportunity to receive DTP1 according to standard immunization schedules, while minimizing recall bias associated with older children Vaccination status was determined using a hierarchical approach based on available documentation. The breakdown of data sources for DTP1 vaccination status determination was as follows: based on vaccination cards with recorded dates, based on vaccination marked on cards without dates, based on maternal recall when cards were unavailable, and recorded as “don’t know” and treated as missing data.

### 2.8. Explanatory Variables

The explanatory variables were selected based on the Three Delays Model framework and existing literature demonstrating their association with childhood vaccination patterns (Table A1). Variables were categorized into individual-, community-, and country-level factors, representing the hierarchical nature of influences on vaccination outcomes.

### 2.9. Individual-Level Variables

Maternal characteristics included age (measured in years), education level (categorized as no education, primary, secondary, or higher education), and parity (total children ever born). Household socioeconomic status was measured using the DHS wealth index, which employs principal component analysis of household assets and living conditions to create quintiles from poorest to richest. Partner’s education level was categorized similarly to maternal education (no education, primary, secondary, or higher).

Maternal empowerment was captured through multiple indicators reflecting women’s autonomy and decision-making capacity. Decision-making power was assessed based on women’s participation in household decisions regarding healthcare, major purchases, and family visits. Media access was measured as exposure to newspapers, radio, or television, representing access to health information. Employment status was dichotomized as working versus not working outside the home.

A composite maternal health-seeking behavior index was constructed to capture patterns of healthcare utilization during pregnancy and delivery. This index incorporated four key components: (1) possession of a health card or vaccination record (hcard), (2) receipt of antenatal care (ante), (3) delivery at a health facility (hospital), and (4) receipt of tetanus vaccination during pregnancy (tetanus). The index was created by summing the number of positive health-seeking behaviors, ranging from 0 (no health-seeking behaviors) to 4 (all health-seeking behaviors present). For analytical purposes, this was further categorized into poor/no health-seeking behavior (0–2 behaviors) versus adequate health-seeking behavior (3–4 behaviors).

Healthcare access barriers were measured through self-reported problems with money and distance in accessing healthcare services. Health insurance coverage was assessed as a binary variable indicating presence or absence of any health insurance. Household size was included as a continuous variable representing the number of household members.

### 2.10. Community-Level Variables

Place of residence was categorized as urban versus rural based on DHS classification. Community-level socioeconomic indicators were calculated as the proportion of individuals within each PSU exhibiting specific characteristics. Community poverty rate represented the proportion of households in the poorest wealth quintile within each community. Community illiteracy rate was calculated as the proportion of women with no formal education within each community. Community unemployment rate represented the proportion of women not engaged in paid employment within each community. These community-level indicators were constructed by aggregating individual responses within PSUs, providing measures of neighborhood socioeconomic context that may influence vaccination behaviors through social norms, peer effects, and local resource availability.

### 2.11. Country-Level Variables

Country-level variables encompassed health system characteristics and broader development indicators. Health system indicators included percentage of GDP spent on health expenditure and government health expenditure as a proportion of total health expenditure. Development measures included the Human Development Index (HDI), Gender Development Index (GDI), and Gender Inequality Index (GII), obtained from the United Nations Development Programme. For analytical purposes, continuous country-level variables were dichotomized using median splits to create low versus high categories, facilitating interpretation of results in policy-relevant terms. This approach allows for examination of whether countries above or below median performance levels show different patterns of zero-dose prevalence.

Temporal variables included survey year and child’s birth year to control for time trends and potential cohort effects in vaccination coverage. These variables account for improvements in vaccination programs over time and ensure that associations reflect contemporaneous relationships rather than temporal changes in service availability or policies.

### 2.12. Bias

Several potential sources of bias were identified and addressed. Selection bias was minimized through the DHS standardized multi-stage sampling methodology designed to achieve nationally representative samples. Information bias related to vaccination status was addressed by prioritizing vaccination card documentation over maternal recall, though recall bias remains possible for children without cards. To assess this, we will report the proportion of vaccination status determinations based on cards versus recall. Confounding bias was addressed through multilevel modeling that simultaneously controls for individual-, community-, and country-level factors. Clustering bias was explicitly modeled through random effects at community and country levels. Non-response bias was minimized through DHS standardized weighting procedures that adjust for differential response rates across population subgroups.

### 2.13. Study Size

The sample size of 30,500 children across 28 countries was determined by DHS sampling procedures designed to provide reliable estimates at national and subnational levels. For multilevel analysis, this sample provides adequate power (>80%) to detect odds ratios of 1.5 or greater for individual-level factors and 2.0 or greater for community- and country-level factors, assuming alpha of 0.05 and accounting for clustering effects. The hierarchical structure (30,500 children nested within 15,637 communities across 28 countries) meets recommended guidelines for multilevel modeling of at least 20 units per level and sufficient variation in outcome prevalence across higher-level units.

### 2.14. Quantitative Variables

All continuous variables were assessed for normality and appropriate transformations applied where necessary. Community-level variables were calculated as proportions within primary sampling units and treated as continuous measures ranging from 0 to 1. Country-level variables were standardized and dichotomized using median splits to facilitate interpretation and policy relevance. Missing data were handled through multiple imputation for individual-level variables with <5% missingness, whilst cases with missing outcome data were excluded from analysis. Sampling weights provided by DHS were applied to all analyses to ensure nationally representative estimates and account for complex survey design, stratification, and clustering effects.

### 2.15. Statistical Methods

#### Descriptive Statistics

Analysis began with pooling datasets from the 28 sub-Saharan African countries to create a single comprehensive dataset for regional analysis. The pooled dataset was weighted using DHS-provided sampling weights to adjust for complex survey design, including stratification, clustering, and differential sampling probabilities across countries. These weights ensure that estimates are representative of the target populations in each country and appropriately account for the multi-stage sampling design.

Descriptive statistics comprised frequency distributions and percentages for all categorical variables, and means with standard deviations for continuous variables, stratified by zero-dose status. To examine the distribution of zero-dose children across the three-level hierarchy, we calculated the number and percentage of zero-dose children at each level: 30,500 children (level 1) nested within 15,637 communities (level 2) across 28 countries (level 3).

Bivariate associations between each explanatory variable and zero-dose status were assessed using Chi-square tests for categorical variables and t-tests for continuous variables. Forest plots were constructed to visualize the variation in zero-dose prevalence across countries, with random-effects meta-analysis used to calculate pooled estimates and assess between-country heterogeneity using the I^2^ statistic. All variables demonstrating significant associations (*p* < 0.05) in bivariate analysis were included in subsequent multilevel modeling.

### 2.16. Modeling Approaches

We employed multivariable logistic multilevel regression models to analyze the association between individual compositional and community and country contextual factors associated with zero-dose status. We specified a three-level model for the binary outcome of zero-dose status for children (level 1), nested within communities (level 2), living in countries (level 3).

We constructed five nested multilevel models to examine the determinants of zero-dose status across different levels of influence. The first model (Model 1) was an unconditional model without any explanatory variables, specified to decompose the amount of variance that existed between country and community levels and establish baseline clustering effects. The second model (Model 2) included only individual-level factors to assess their independent contribution. The third model (Model 3) contained only community-level factors to examine contextual effects. The fourth model (Model 4) included only country-level factors to assess macro-level influences. Finally, the fifth model (Model 5) simultaneously controlled for individual-, community-, and country-level factors, representing the full model.

This sequential modeling approach allows for assessment of how much variance at each level is explained by different categories of factors and enables examination of how contextual effects change when individual characteristics are controlled.

### 2.17. Fixed Effects (Measures of Association)

The results of fixed effects (measures of association) were reported as odds ratios (ORs) with their 95% credible intervals (CrIs). Bayesian statistical inference provides probability distributions for measures of association, which are summarized with 95% credible intervals rather than traditional 95% confidence intervals. A 95% credible interval can be interpreted as there being a 95% probability that the parameter takes a value in the specified range.

For categorical variables, odds ratios represent the odds of zero-dose status in each category relative to the reference category. For continuous variables, odds ratios represent the change in odds associated with a one-unit increase in the predictor variable. Statistical significance was assessed based on whether the 95% credible interval excluded 1.0, indicating evidence of association with zero-dose status.

### 2.18. Random Effects (Measures of Variation)

The contextual effects were measured using several indicators of clustering and variation. The variance partition coefficient (VPC) was calculated to measure similarity between children in the same community and within the same country. The VPC represents the percentage of total variance in the probability of zero-dose status that is attributable to community- and country-level factors, quantifying the degree of clustering of vaccination behaviors within these geographic units.

The median odds ratio (MOR) was calculated to quantify the extent of variation between clusters. The MOR represents the median value of the odds ratio between a child at higher risk and a child at lower risk when randomly selecting two children from different communities or countries. MOR values close to 1.0 indicate minimal clustering, while larger values indicate substantial variation between areas.

The proportional change in variance (PCV) was calculated to assess the proportion of area-level variance explained by including explanatory variables in the model, calculated as: PCV = (σ^2^-null—σ^2^-model)/σ^2^-null × 100%, where σ^2^-null is the area-level variance in the null model and σ^2^-model is the area-level variance in the model with covariates.

### 2.19. Model Fit and Specifications

All analyses were conducted using MLwiN software version 3.13, with parameters estimated using the Markov Chain Monte Carlo (MCMC) procedure. MCMC estimation was chosen over traditional maximum likelihood methods due to its superior performance with complex multilevel models and its ability to provide full Bayesian inference, including credible intervals for all parameters.

The MCMC chains were run for 50,000 iterations following a burn-in period of 5000 iterations to ensure convergence. Convergence was assessed through visual inspection of trace plots and calculation of effective sample sizes. Multiple chains were run with different starting values to confirm consistent results across runs.

The Bayesian Deviance Information Criterion (DIC) was used to assess model fit and compare models, with lower DIC values indicating better fit to the data. The DIC balances model fit with model complexity, penalizing models with excessive parameters. Model selection was based on both statistical criteria (DIC values) and theoretical considerations regarding the relevance of included variables.

This study was conducted and reported in accordance with the STROBE (Strengthening the Reporting of Observational Studies in Epidemiology) guidelines for cross-sectional studies [19] (Table A2).

### 2.20. Ethical Considerations

Our study is a secondary analysis of existing DHS data collected by the DHS Program. The surveys were conducted in accordance with the principles outlined in the Declaration of Helsinki (1975, revised in 2013). The survey procedures and questionnaires were approved by the ICF Institutional Review Board in the United States of America and by national ethics committees in all the 28 countries where the surveys were conducted. All study participants gave informed consent before participation, and all information was collected confidentially. We obtained the raw survey data and written consent of the DHS Program to use the data for research purposes. More information on ethical review and approval in DHSs is freely available from https://www.dhsprogram.com/Methodology/Protecting-the-Privacy-of-DHS-Survey-Respondents.cfm (accessed on 18 August 2025).

## 3. Results

### 3.1. Sample Characteristics

The pooled analysis included 30,500 children aged 12–23 months from Demographic and Health Surveys conducted across sub-Saharan African countries between 2015 and 2024 (Figure 1). Of these children, 3718 (12.2%) were classified as zero-dose children who had not received their first dose of DTP vaccine (Table 1).

Maternal and household characteristics showed notable disparities between zero-dose and vaccinated children. Mothers of zero-dose children had substantially lower educational attainment, with 56.2% having no formal education compared to 24.6% among mothers of vaccinated children. Similarly, partners of mothers with zero-dose children demonstrated lower educational levels, with 55.3% having no education versus 25.3% in the comparison group.

Healthcare utilization patterns differed markedly between groups. A substantial proportion of mothers with zero-dose children (26.2%) had received no antenatal care visits compared to only 0.7% of mothers whose children were vaccinated. Among those who did receive antenatal care, mothers of zero-dose children were less likely to complete the recommended four or more visits (16.3% vs. 60.1%). Examination of parity patterns revealed an increasing proportion of zero-dose children with higher birth orders, particularly from parity 5 onwards, suggesting that vaccination-seeking behavior may be influenced by competing demands in larger families.

Barriers to healthcare access were more prevalent among families with zero-dose children. Financial constraints were reported by 55.2% of families with zero-dose children compared to 45.2% of those with vaccinated children. Geographic barriers were similarly pronounced, with 44.2% of zero-dose families reporting distance as a problem accessing care versus 31.7% in the comparison group.

Community and country-level factors revealed systematic disadvantages for zero-dose children. These children were predominantly from rural areas (73.3% vs. 60.7%) and resided in communities with higher poverty rates (30.8% vs. 18.2%) and illiteracy rates (54.7% vs. 27.8%). At the country level, zero-dose children lived in nations with slightly higher GDP per capita (4525.8 vs. 4075.6 USD) but lower health expenditure as a percentage of GDP (4.4% vs. 5.1%). Human development indicators, including the Human Development Index, Gender Development Index, and Gender Inequality Index, showed minimal variation between groups.

### 3.2. Variation in Zero-Dose Prevalence Across Sub-Saharan African Countries

The prevalence of zero-dose children varied substantially across the 28 sub-Saharan African countries included in the analysis (Figure 2). This marked heterogeneity was statistically significant across all regional groupings (I^2^ > 93% for all regions, *p* < 0.001), indicating considerable between-country variation that extends beyond sampling variation.

Central Africa demonstrated the highest regional burden, with a pooled prevalence of 24.13% (95% CI: 13.41–36.82). Within this region, Chad exhibited the highest zero-dose prevalence at 40.00%, followed by Angola at 29.38%. Gabon and Central African Republic showed more moderate rates at 15.87 and 14.14%, respectively.

West Africa showed considerable internal variation with a pooled prevalence of 10.83% (95% CI: 5.36–17.90). Guinea recorded the highest prevalence in this region at 33.41% (95% CI: 33.54–38.43), followed by Nigeria at 28.63% (95% CI: 31.32–33.60) and Côte d’Ivoire at 23.43% (95% CI: 22.49–26.21).

East Africa presented more heterogeneous patterns with a pooled prevalence of 5.20% (95% CI: 2.37–9.03). Ethiopia and Madagascar showed the highest burdens at 20.14% and 16.79%, respectively. Several East African countries demonstrated remarkably low prevalences, including Rwanda at 0.51%, Burundi at 1.45%, and Zambia at 1.76%.

Southern Africa exhibited the lowest regional burden with a pooled prevalence of 3.01% (95% CI: 0.40–7.76), though this estimate was based on only two countries. South Africa showed a prevalence of 5.17%, while Lesotho demonstrated 1.38%.

### 3.3. Measures of Association (Fixed Effects Model)

The fully adjusted multilevel model (Model 5) incorporating individual-, community-, and country-level factors revealed several significant associations with zero-dose status among children aged 12–23 months in sub-Saharan Africa (Table 2).

Individual-level factors demonstrated the strongest associations with zero-dose status. The most pronounced effect was observed for poor or absent maternal health-seeking behavior, with children whose mothers had inadequate antenatal care utilization showing dramatically increased odds of being zero-dose (OR = 12.00, 95% CrI: 9.78–14.55).

Partner education also emerged as a significant determinant, with children whose fathers/partners had no formal education showing increased odds of zero-dose status (OR = 1.52, 95% CrI: 1.20–1.96). Maternal empowerment indicators were consistently associated with vaccination outcomes, as children of mothers with no decision-making power had 23% higher odds of being zero-dose (OR = 1.23, 95% CrI: 1.08–1.39), while those with no media access showed 34% higher odds (OR = 1.34, 95% CrI: 1.18–1.48). In addition, children of women who reported money problem accessing care (OR = 1.98, 95% CrI 1.77–2.22) had higher odds of being zero-dose.

Community illiteracy rates demonstrated a significant positive association, with each unit increase in illiteracy rate associated with 8% higher odds of zero-dose status (OR = 1.08, 95% CrI: 1.05–1.11). Similarly, community unemployment rates showed a modest but significant association, with each unit increase associated with 5% higher odds of being zero-dose (OR = 1.05, 95% CrI: 1.02–1.08).

Children in countries with low health expenditure (below median) had substantially higher odds of zero-dose status compared to those in higher health expenditure countries (OR = 2.29, 95% CrI: 1.31–3.96), indicating that inadequate national health investment is associated with poorer vaccination coverage.

### 3.4. Measures of Variations (Random Effects)

As shown in Table 2, in Model 1 (unconditional model), there was significant variation in the odds of zero-dose status across countries (σ^2^ = 2.65, 95% CrI 1.52 to 4.63) and across communities (σ^2^ = 2.40, 95% CrI 2.10 to 2.79). According to the intra-country and intra-community correlation coefficients, 31.8% and 60.5% of the variance in odds of zero-dose status could be attributed to country- and community-level factors, respectively.

From the full model (Model 5), which included individual-, community-, and country-level factors, the total explained variance was 71.1% at the country level and 98.7% at the community level. From the full model (Model 5), it was estimated that if a child moved to another community or another country with a higher probability of zero-dose status, the median increase in their odds of being zero-dose would be 2.30 (95% CrI 1.84 to 3.08) and 1.19-fold (95% CrI 1.12 to 1.27), respectively. These substantial MOR values underscore the important role of contextual factors in shaping vaccination coverage patterns beyond individual-level characteristics.

## 4. Discussion

### 4.1. Main Findings

This multilevel analysis of 30,500 children from 28 sub-Saharan African countries identified a 12.19% zero-dose prevalence with substantial variation across countries (ranging from 0.51% in Rwanda to 40.00% in Chad). The Three Delays Model effectively identified critical barriers, with maternal health-seeking behavior emerging as the strongest predictor (OR = 12.00, 95% CrI: 9.78–14.55).

Individual-level factors dominated, particularly educational gradients where paternal lack of education increased odds 1.52-fold, and maternal empowerment deficits including financial barriers (OR = 1.98) and limited decision-making power (OR = 1.23). Community-level illiteracy rates showed 8% increased odds per unit increase, whilst country-level low health expenditure demonstrated 2.3-fold higher odds. The observed association between higher parity and increased zero-dose prevalence warrants particular attention, as it suggests that mothers with larger families may face greater challenges in prioritizing vaccination among competing household demands and resource constraints.

Substantial geographical clustering persisted, with 19.5% of variance at community level and 18.7% at country level in the final model. Sequential modeling revealed individual factors explained the largest proportion of variance, followed by community and country factors. This pattern indicates that whilst contextual factors matter, maternal and household characteristics remain primary drivers, necessitating interventions addressing both individual and structural determinants simultaneously.

### 4.2. Comparison with Previous Studies

Our findings align with recent studies examining zero-dose children in sub-Saharan Africa whilst revealing important regional variations. The overall zero-dose prevalence of 12.19% observed in our study is consistent with country-specific analyses from the region, though individual country estimates show substantial variation. Recent studies from

Ethiopia reported zero-dose prevalences ranging from 16.3% to 23.7% [20,21], whilst Tanzania demonstrated a lower prevalence of 7.45% [22]. Our finding of 0.51% in Rwanda aligns with previous recognition of Rwanda’s exceptional vaccination performance, whilst the 40.00% prevalence in Chad reflects the persistent challenges in Central African settings.

Furthermore, our findings corroborate recent international research on zero-dose children, though important regional variations emerge across the study countries. Our zero-dose prevalence of 12.19% falls within the range reported by Cata-Preta et al. (2021) in their comprehensive analysis of 92 low- and middle-income countries, which found an overall zero-dose prevalence of 7.7% using the four basic vaccines definition, with considerable variation across countries [23]. More recently, studies from India have documented substantial progress in reducing zero-dose children, with Johri et al. (2021) reporting a decline from 33.4% in 1992 to 10.1% in 2016 [24]. Latin American countries present a mixed picture, with recent analyses by Ávila-Agüero et al. (2025) [25] documenting approximately 2.7 million under-vaccinated infants across the region, with DTP1 coverage ranging from 65% in Venezuela to 99% in the Dominican Republic, Chile, Cuba, and Costa Rica [25]. Collectively, these comparisons extend the relevance of our findings and the applied Three Delays Model beyond sub-Saharan Africa. They underscore the universal challenge of reaching zero-dose children and the imperative for contextually nuanced, equity-oriented vaccination strategies tailored to local social, cultural, and health system realities.

The community-level clustering effects we observed are consistent with recent spatial analyses. Muchie et al. (2025 identified similar geographic clustering in Ethiopia, with hotspots concentrated in southwest and northeast regions) [21]. Similarly, Gichuki et al. (2025) demonstrated distinct spatial patterns in Kenya, with northern regions showing elevated zero-dose prevalence [26]. These findings support our observation that 19.5% of variance occurs at community level, indicating that place-based factors create shared environments influencing vaccination behaviors.

The overwhelming importance of maternal health-seeking behavior (OR = 12.00) aligns with recent evidence from across the region. Studies from Ethiopia consistently identified antenatal care utilization as a critical predictor, with Bogale et al. (2025) reporting that facility delivery increased vaccination likelihood 1.57-fold [27]. Similarly, Mohamoud et al. (2024) [28] found that Somali mothers attending antenatal care had substantially reduced odds of having zero-dose children (OR = 0.161). This consistency across diverse contexts reinforces the critical pathway from maternal healthcare engagement to childhood vaccination.

Recent studies have confirmed the persistent rural-urban disparities we observed, though the mechanisms differ across contexts. Santos et al. (2024) demonstrated that whilst urban children generally show vaccination advantages, poor urban children face unique challenges, with zero-dose prevalence of 12.6% among urban poor compared to 6.5% among urban non-poor across 97 countries [29]. However, our finding that rural residence showed no independent association after controlling for socioeconomic factors suggests that geographic barriers may be increasingly mediated through educational and economic pathways rather than distance alone [30].

### 4.3. Implications for Policy and Future Research

Our findings provide clear evidence-based guidance for reducing zero-dose children in sub-Saharan Africa. The twelve-fold increased odds among children whose mothers demonstrate poor health-seeking behavior demands immediate integration of antenatal care platforms with childhood vaccination services. Countries should establish systematic linkages ensuring every antenatal visit includes vaccination counseling and appointment scheduling [31,32].

The pronounced educational gradients, particularly the 52% increased odds among children of uneducated fathers, necessitate targeted interventions for low-literacy households. Community health worker programs should prioritize families with limited parental education, providing tailored navigation support and health literacy interventions [33,34,35,36].

Community-level determinants require place-based approaches. The 8% increased odds per unit increase in community illiteracy rates supports intersectoral collaboration between education and health ministries, whilst the financial barriers identified (OR = 1.98) mandate elimination of user fees and transportation support in high-burden areas [37,38,39,40].

The 2.3-fold increased odds in low health expenditure countries provides compelling evidence for increased domestic health investment. Governments must prioritize reaching the Abuja Declaration target of 15% health spending, with specific emphasis on primary healthcare and immunization services [41,42].

The substantial clustering effects (19.5% community, 18.7% country variance) indicate that context-specific adaptations remain essential, suggesting that successful interventions require local tailoring rather than standardized approaches across the region [43,44,45].

Future research priorities should address the substantial unexplained variance at community and country levels through mixed-methods studies that combine quantitative surveys with qualitative investigations of local health system performance, cultural beliefs, and social network influences [46,47].

### 4.4. Study Strengths and Limitations

This study presents several methodological strengths. The analysis utilized a large, representative sample of 30,500 children from 28 sub-Saharan African countries, providing substantial statistical power and broad geographic coverage [48]. Standardized DHS methodology ensures comparability across countries and addresses data harmonization concerns common in multi-country analyses. The multilevel modeling approach appropriately accounts for hierarchical data structure and clustering effects, providing more accurate estimates than single-level analyses [43]. The Three Delays Model provided a structured theoretical framework for systematic examination of vaccination barriers [49], whilst the inclusion of compositional and contextual factors aligns with best practices in health services research [50].

Several limitations must be acknowledged. The cross-sectional design precludes causal inference, requiring longitudinal studies to confirm causal pathways. Vaccination status determination relies partially on maternal recall, potentially introducing information bias, though DHS methodology remains the gold standard for population-based estimates in low-income settings [51]. The reliability of our outcome measure varied by documentation source, with vaccination cards providing the most reliable data and maternal recall potentially subject to recall bias. However, the DHS methodology employs systematic approaches to minimize recall bias, and validation studies have demonstrated reasonable accuracy of maternal recall for first-dose vaccines when cards are unavailable [52]. The analysis was constrained by available DHS variables, potentially omitting important determinants such as healthcare provider characteristics, vaccine stockouts, and cultural beliefs [52]. Substantial residual variance at community (19.5%) and country (18.7%) levels indicate that unmeasured contextual factors remain important. Community-level variable aggregation may not accurately reflect true neighborhood characteristics, whilst median splits for country variables may mask non-linear relationships [50]. The analysis was also limited by the absence of data on mobile phone access and usage, which represents an increasingly important source of health information in sub-Saharan Africa [53,54]. Future studies should examine the role of mobile technology in health information seeking and vaccination decision-making, as this pathway was not captured in the traditional media exposure variables available in the DHSs analyzed [53,54].

### 4.5. Generalisability of Findings

The inclusion of 28 sub-Saharan African countries with standardized DHS methodology enhances the generalizability of our findings across the region. The consistent operationalization of variables using identical survey instruments and protocols across diverse contexts–from post-conflict settings to stable democracies, from least developed to middle-income countries–strengthens the external validity of observed associations.

The marked heterogeneity in zero-dose prevalence (0.51% to 40.00%) demonstrates that whilst the determinants we identified operate consistently across contexts, their relative importance and policy implications may vary by country. This variation supports the generalizability of our multilevel framework whilst emphasizing the need for context-specific adaptations of interventions.

The findings are most directly applicable to other sub-Saharan African countries with similar health system structures and demographic profiles. Caution is warranted when extrapolating to high-income settings or regions with different vaccination delivery systems, though the Three Delays Model framework may remain relevant for understanding barriers in diverse contexts.

## 5. Conclusions

This comprehensive multilevel analysis demonstrates that zero-dose children in sub-Saharan Africa are concentrated among the most disadvantaged populations, with the Three Delays Model successfully identifying critical barriers across healthcare-seeking pathways. The twelve-fold increased odds among children whose mothers demonstrate poor health-seeking behavior provides compelling evidence for immediate integration of childhood vaccination services with antenatal care and other primary healthcare services.

The findings necessitate coordinated interventions spanning multiple levels. Countries must immediately eliminate financial barriers to vaccination, target high-parity mothers with tailored services, and address educational gradients through community health literacy programs. The 2.3-fold-increased odds in low health expenditure countries demand urgent progress toward the target of 15% health spending.

## Figures and Tables

**Figure 1 vaccines-13-00987-f001:**
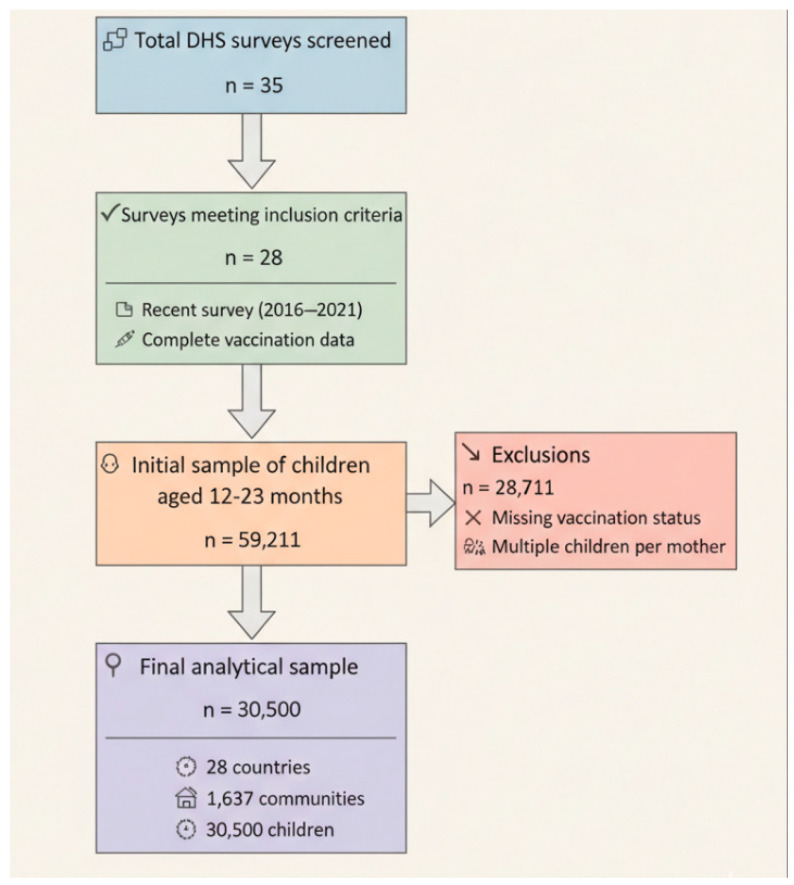
Flowchart of Survey Screening and Sample Selection Process.

**Figure 2 vaccines-13-00987-f002:**
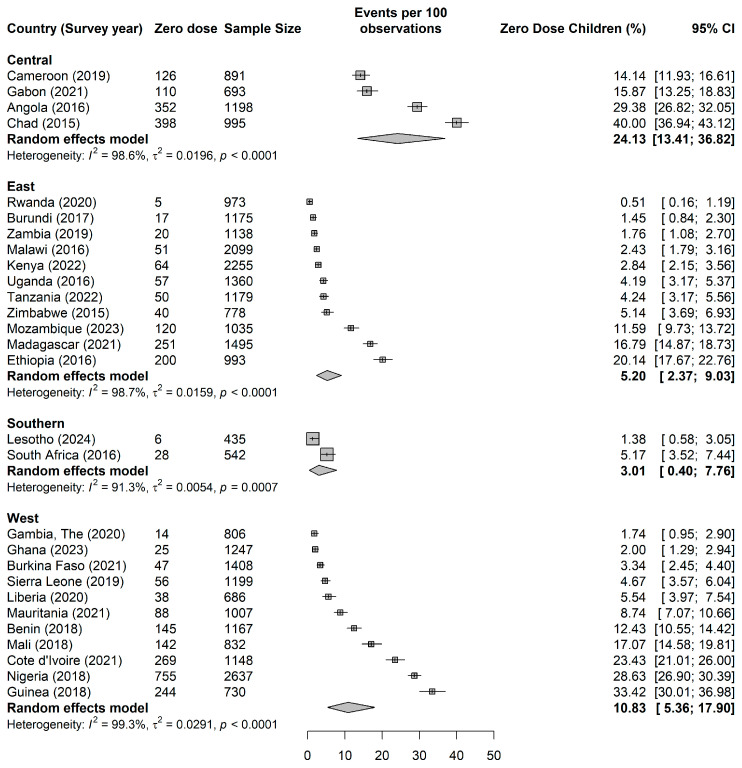
Variations in Zero-dose Children Across Countries. Note: Individual study estimates are shown as squares (■), with the size proportional to study weight. Horizontal lines represent 95% confidence intervals (-). Diamond symbols (◊) represent pooled estimates from random effects meta-analysis for each regional subgroup, with the width of the diamond indicating the 95% confidence interval of the pooled estimate.

**Table 1 vaccines-13-00987-t001:** Summary of pooled sample characteristics of the demographic and health survey data in sub-Saharan Africa.

	Zero Dose		
	No	Yes	Total
N	26,834 (88.0%)	3655 (12.0%)	30,489 (100.0%)
Survey year	2019 (2015 2024)	2019 (2015 2024)	2019 (2015 2024)
Birth year	2017 (2014 2023)	2017 (2014 2023)	2017 (2014 2023)
Maternal age			
Young Adult	10,734 (40.0%)	1634 (44.7%)	12,369 (40.6%)
Adult	14,297 (53.3%)	1722 (47.1%)	16,019 (52.5%)
Middle-Aged/Older Adult	1802 (6.7%)	299 (8.2%)	2101 (6.9%)
Maternal education			
no education	6605 (24.6%)	2054 (56.2%)	8659 (28.4%)
primary	8951 (33.4%)	882 (24.1%)	9833 (32.2%)
secondary	9388 (35.0%)	630 (17.2%)	10,018 (32.9%)
higher	1890 (7.0%)	89 (2.4%)	1979 (6.5%)
Wealth index			
poorest	4623 (17.2%)	1132 (31.0%)	5755 (18.9%)
poorer	5114 (19.1%)	931 (25.5%)	6045 (19.8%)
middle	5409 (20.2%)	707 (19.3%)	6116 (20.1%)
richer	5746 (21.4%)	524 (14.3%)	6271 (20.6%)
richest	5941 (22.1%)	361 (9.9%)	6303 (20.7%)
husband/partner’s education level			
no education	5228 (25.3%)	1616 (55.3%)	6845 (29.0%)
primary	6063 (29.3%)	598 (20.4%)	6661 (28.2%)
secondary	6999 (33.8%)	588 (20.1%)	7587 (32.1%)
higher	2390 (11.6%)	122 (4.2%)	2512 (10.6%)
Antenatal visits			
no visit	1421 (5.5%)	1238 (35.7%)	2659 (9.1%)
lessthan4	6707 (26.1%)	996 (28.7%)	7703 (26.4%)
4 or more	17,563 (68.4%)	1235 (35.6%)	18,798 (64.5%)
Parity			
1	11,426 (42.6%)	1502 (41.1%)	12,928 (42.4%)
2	4448 (16.6%)	495 (13.5%)	4943 (16.2%)
3	3476 (13.0%)	417 (11.4%)	3893 (12.8%)
4	2613 (9.7%)	319 (8.7%)	2933 (9.6%)
5	1865 (7.0%)	316 (8.6%)	2181 (7.2%)
6	1288 (4.8%)	195 (5.3%)	1482 (4.9%)
7	806 (3.0%)	160 (4.4%)	966 (3.2%)
8	467 (1.7%)	102 (2.8%)	569 (1.9%)
9	258 (1.0%)	62 (1.7%)	320 (1.0%)
10	186 (0.7%)	87 (2.4%)	273 (0.9%)
Not working			
0	16,198 (60.4%)	1936 (53.0%)	18,134 (59.5%)
1	10,636 (39.6%)	1719 (47.0%)	12,355 (40.5%)
No decision-making power			
0	16,797 (62.6%)	1881 (51.5%)	18,677 (61.3%)
1	10,037 (37.4%)	1775 (48.5%)	11,812 (38.7%)
No media access			
0	18,828 (70.2%)	1679 (45.9%)	20,506 (67.3%)
1	8006 (29.8%)	1976 (54.1%)	9982 (32.7%)
Household size			
1	69 (0.3%)	8 (0.2%)	77 (0.3%)
2	723 (2.7%)	116 (3.2%)	839 (2.8%)
3	5192 (19.3%)	680 (18.6%)	5872 (19.3%)
4	4167 (15.5%)	496 (13.6%)	4663 (15.3%)
5	4016 (15.0%)	541 (14.8%)	4557 (14.9%)
6	3411 (12.7%)	385 (10.5%)	3796 (12.5%)
7	2578 (9.6%)	379 (10.4%)	2957 (9.7%)
8	1861 (6.9%)	271 (7.4%)	2132 (7.0%)
9	1247 (4.6%)	187 (5.1%)	1434 (4.7%)
10	3570 (13.3%)	591 (16.2%)	4161 (13.6%)
Money problem accessing care			
No	14,700 (54.8%)	1639 (44.8%)	16,339 (53.6%)
Yes	12,133 (45.2%)	2017 (55.2%)	14,150 (46.4%)
Distance problem accessing care			
No	18,321 (68.3%)	2038 (55.8%)	20,359 (66.8%)
Yes	8513 (31.7%)	1617 (44.2%)	10,130 (33.2%)
Health insurance			
No	6055 (22.6%)	492 (13.5%)	6546 (21.5%)
Yes	20,779 (77.4%)	3163 (86.5%)	23,942 (78.5%)
Place of resident			
Urban	10,533 (39.3%)	967 (26.4%)	11,499 (37.7%)
Rural	16,301 (60.7%)	2689 (73.6%)	18,990 (62.3%)
Community poverty rate	18.2 (24.9)	28.9 (30.8)	19.5 (25.9)
Community illiteracy rate	27.8 (29.7)	54.7 (34.1)	31.0 (31.5)
Community unemployment rate	31.0 (26.1)	31.3 (30.3)	31.0 (26.6)
Gross domestic product	4075.6 (3270.2)	4525.8 (3384.7)	4129.6 (3287.4)
Percentage health expenditure	5.1 (2.0)	4.4 (1.4)	5.1 (1.9)
Human development index	0.5 (0.1)	0.5 (0.1)	0.5 (0.1)
Gender Development Index	0.9 (0.0)	0.9 (0.0)	0.9 (0.0)
Gender Inequality Index	0.6 (0.1)	0.6 (0.1)	0.6 (0.1)

**Table 2 vaccines-13-00987-t002:** Individual compositional and contextual factors associated with zero-dose children.

	Model 1	Model 2	Model 3	Model 4	Model 5
	OR (95% CrI)	OR (95% CrI)	OR (95% CrI)	OR (95% CrI)	OR (95% CrI)
Measures of associations (Fixed Effects Model)					
Individual-level factors					
Survey year		1.02 (1.01–1.02)			1.03 (1.03–1.03)
Birth year		0.91 (0.91–0.91)			0.92 (0.91–0.92)
Maternal age					
Young adult		1.40 (1.08–1.82)			1.35 (1.04–1.69)
Adult		1.00 (0.82–1.22)			0.99 (0.80–1.20)
Middle-Age/Older Adult					
Education					
No education		1.92 (1.32–2.75)			1.51 (0.97–2.17)
Primary		1.32 (0.90–1.89)			1.24 (0.81–1.72)
Secondary		1.13 (0.79–1.61)			1.11 (0.73–1.54)
Tertiary					
Wealth					
Poorest		1.29 (1.01–1.54)			1.22 (0.94–1.55)
Poorer		1.15 (0.92–1.38)			1.13 (0.90–1.39)
Middle		1.11 (0.90–1.33)			1.09 (0.89–1.33)
Richer		0.95 (0.77–1.13)			0.94 (0.77–1.15)
Richest					
Partner Education					
No education		1.66 (1.31–2.05)			1.52 (1.20–1.96)
Primary		1.22 (0.96–1.52)			1.26 (0.98–1.65)
Secondary		1.12 (0.90–1.38)			1.17 (0.92–1.51)
Tertiary					
Parity		1.01 (0.98–1.04)			1.02 (0.98–1.04)
Not working		1.20 (1.06–1.34)			1.09 (0.97–1.23)
No decision-making power		1.27 (1.13–1.41)			1.23 (1.08–1.39)
No media access		1.34 (1.19–1.49)			1.32 (1.18–1.48)
Household size		1.00 (0.98–1.02)			1.00 (0.98–1.02)
Money problem accessing care		0.95 (0.84–1.07)			0.96 (0.84–1.08)
Distance problem accessing care		1.16 (1.02–1.30)			1.14 (1.00–1.29)
Money problem accessing care		2.05 (1.83–2.31)			1.98 (1.77–2.22)
No health insurance		1.10 (0.87–1.36)			1.13 (0.92–1.41)
Poor/No maternal health seeking		12.62 (10.37–15.29)			12.00 (9.78–14.55)
Community -level factors					
Rural resident			1.27 (1.12–1.46)		0.93 (0.80–1.07)
Community poverty rate			1.07 (1.04–1.09)		1.00 (0.98–1.03)
Community illiteracy rate			1.30 (1.27–1.33)		1.08 (1.05–1.11)
Community unemployment rate			1.10 (1.07–1.12)		1.05 (1.02–1.08)
Societal -level factors					
Gross domestic product				0.74 (0.22–1.61)	0.63 (0.26–1.29)
Percentage health expenditure				1.12 (0.54–2.17)	2.29 (1.31–3.96)
Human development index				0.70 (0.34–1.66)	1.51 (0.64–2.57)
Gender Development Index				6.83 (3.28–12.75)	1.52 (0.89–2.38)
Gender Inequality Index				0.81 (0.30–1.60)	1.22 (0.68–2.21)
Measures of variations (random effects)					
Country-level					
Variance (95% CrI)	2.65 (1.52–4.63)	1.15 (0.64–2.00)	1.86 (1.05–3.27)	1.91 (1.03–3.46)	0.77 (0.41–1.39)
VPC (%)	31.8 (22.0–43.2)	25.9 (16.6–37.8)	28.1 (18.8–39.5)	25.0–15.9–36.0)	18.7 (11.0–29.3)
MOR (95% CrI)	4.72 (3.24–7.79)	2.78 (2.14–3.85)	3.68 (2.66–5.62)	3.74 (2.63–5.89)	2.30 (1.84–3.08)
Explained variance (%)	reference	56.6 (56.8–57.9)	29.7 (29.3–30.7)	27.8 (25.3–32.5)	71.1 (70.0–73.1)
Community-level					
Variance (95% CrI)	2.40 (2.10–2.79)	0.00 (0.00–0.00)	1.47 (1.25–1.72)	2.45 (2.12–2.85)	0.03 (0.01–0.06)
VPC (%)	60.5 (52.4–69.3)	25.9 (16.6–37.8)	50.3 (−41.2–60.2)	57.0 (48.9–65.7)	19.5 (11.4–30.6)
MOR (95% CrI)	4.38 (3.98–4.92)	1.04 (1.03–1.05)	3.18 (2.91–3.49)	4.45 (4.02–5.00)	1.19 (1.12–1.27)
Explained variance (%)	reference	99.9 (99.9–99.9)	38.6 (38.5–40.3)	−1.9 (−2.2–−1.1)	98.7 (97.8–99.3)

OR odds ratio, CrI credible interval, MOR median odds ratio, VPC variance partition coefficient. Statistically significant estimates with 95% credible intervals excluding the null value (1.0) are underlined. Model 1—baseline model without any explanatory variables (unconditional model). Model 2—adjusted for only individual-level factors. Model 3—adjusted for only community-level factors. Model 4—adjusted for only country-level factors. Model 5—adjusted for individual-, community-, and country-level factors (full model).

## Data Availability

The datasets analyzed during the current study are available from the DHS Program repository (www.dhsprogram.com). Access to DHS datasets requires free registration and agreement to data use terms.

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
