# Peer review of "Multilevel Analysis of Zero-Dose Children in Sub-Saharan Africa: A Three Delays Model Study"

_vaccines, 2025, doi:10.3390/vaccines13090987_

Round 1

Reviewer 1 Report

Comments and Suggestions for Authors

Dear Authors, the manuscript is interest but several improvements required: - in all part - except for the introduction, which instead requires some further elaboration (sse under) - necessity to consider reduction because redundant and always not full scientific. Title: isn't full clear for the reader. I suggest classical formula of the study conducted and setting and reduce max in 15 words. Abstract: I suggest to extend possible clinical implication of data finding in the conclusion and reduce in classical format of max 250 words. Keywords: I suggest max 4/e words that include the tyoe of stuyd conducted and the setting as the title. Introduction: epidemiological data ok. In this part could be help the reasearch (in clinical practice and Public Health view) to extend the discussion with relevant references of the topic fundamental for international overview as relevant campain of vaccination as "Papilloma vaccination Programs and Knowledge Gaps as Barriers to Implementation" and "Acceptability of human papillomavirus vaccination" that complete the preliminary discussion and extend the discussion in your text extremaly generic in the part that starting with "Traditional approaches.." and ends with "...vaccination patterns". This elements, as the methods suggestions, is fundamental for possible international consideration. The objective (I suppose last paragraph of the study) in not clear; please adopt classical formula as "The priamy aim was... while the secondary was/were..."; Methods: the part that certly required major attention. Not clear the method adopetd and the type of the study conducted. I suggest to adopt international reporting tool (with the references and the check list in the supplementary files) mandatary for scientific community and could help the authors to present the manuscript in international and scientific perspective (e.g. Equator Network: https://www.equator-network.org/). The correction of this elements, as the previous for the introduction, is funtamental for possible international consideratin; • Results: editing improvements suggested for the tables andfigures presented. Can you think insert posible flow chart of the study? Discussion; I suggest to extend in clinical and Public Health review the possible discussion and remember to reduce the section because is very redundant. I also suggest possible insertion of section titled "Implications for..." the elements suggested in the introduction and link with data finding (always attention the lenght). Limits: I suggest to clarify the possible generalization of data finding and reduce alway the section that in some part result redudant. Conclusion: according the previous suggestion, poned the corrections in line with the indications. References: according the previous comments in clinical and public health review and in the methods, extend the references with the suggestions poned and consider to update the references over 10 yera if aren't relevant for high impact evidence based or for the methods framework. Please answer poin-point, mainly for clinical practice implications and methods for possible future considerations.

Comments on the Quality of English Language

Native review recommended

Author Response

Dear Authors, the manuscript is interest but several improvements required: - in all part - except for the introduction, which instead requires some further elaboration (sse under) - necessity to consider reduction because redundant and always not full scientific.

Authors’ reply: We thank the reviewer for their interest in our manuscript and for acknowledging that the topic is of interest. We have carefully considered each of the reviewer's suggestions and have undertaken substantial revisions to address their concerns. We have systematically addressed each specific point raised by the reviewer, including title revision, abstract restructuring, enhanced introduction with relevant references, methodological clarifications using established reporting guidelines, and overall length reduction whilst maintaining scientific completeness.

Below, we provide detailed responses to each specific comment and describe the corresponding revisions made to improve the manuscript's quality and adherence to international publishing standards.

Title: isn't full clear for the reader. I suggest classical formula of the study conducted and setting and reduce max in 15 words.

Authors’ reply: We acknowledge the reviewer's concern about title clarity and length. The reviewer's suggestion to follow a classical formula and limit the title to 15 words is noted, and we have limited it to 16 words now:

“Multilevel Analysis of Zero-Dose Children and Vaccination Barriers in Sub-Saharan Africa: A Three Delays Model Study”.

Abstract: I suggest to extend possible clinical implication of data finding in the conclusion and reduce in classical format of max 250 words.

Authors’ reply: We agree with the reviewer's suggestion to enhance the clinical implications in the abstract conclusion and adhere to a 250-word limit. We have revised both the abstract and main text conclusions to be more concise whilst strengthening the policy implications.

Revised Abstract (207 words):

Background: Zero-dose children represent a critical challenge for achieving universal immunisation coverage in sub-Saharan Africa. This study applies the Three Delays Model to examine multilevel factors associated with zero-dose children.

Methods: We analysed data from 30,500 children aged 12-23 months across 28 sub-Saharan African countries using demographic and health surveys (2015-2024). Zero-dose status was defined as not receiving the first dose of diphtheria-tetanus-pertussis containing vaccine. Multilevel logistic regression models examined individual-, community-, and country-level determinants.

Results: Overall zero-dose prevalence was 12.19% (95% confidence interval 11.82-12.56), ranging from 0.51% in Rwanda to 40.00% in Chad. Poor maternal health-seeking behaviour showed the strongest association: odds ratio (OR) 12.00, 95% credible interval (CrI) 9.78-14.55. Paternal education demonstrated clear gradients, with no formal education increasing odds 1.52-fold. Maternal empowerment factors were significant: lack of decision-making power (OR=1.23), financial barriers (OR=1.98), and no media access (OR=1.32). Community illiteracy rates and low country-level health expenditure were associated with increased zero-dose prevalence. Substantial clustering persisted at community (19.5%) and country (18.7%) levels.

Conclusions: Zero-dose children concentrate among the most disadvantaged populations, with maternal health-seeking behaviour as the strongest predictor. Immediate policy actions should integrate antenatal care with vaccination services, target high-parity mothers, eliminate financial barriers, and increase health expenditure to 15% of national budgets.

Revised Main Text Conclusion:

This comprehensive multilevel analysis demonstrates that zero-dose children in sub-Saharan Africa are concentrated among the most disadvantaged populations, with the Three Delays Model successfully identifying critical barriers across healthcare-seeking pathways. The twelve-fold increased odds among children whose mothers demonstrate poor health-seeking behaviour provides compelling evidence for immediate integration of antenatal care platforms with childhood vaccination services.

The findings necessitate coordinated interventions spanning multiple levels. Countries must immediately eliminate financial barriers to vaccination, target high-parity mothers with tailored services, and address educational gradients through community health literacy programmes. The 2.3-fold increased odds in low health expenditure countries demands urgent progress toward the target of 15% health spending.

Keywords: I suggest max 4/e words that include the type of study conducted and the setting as the title.

Authors’ reply: We acknowledge the reviewer's suggestion to limit keywords to 4-5 words that include the study type and setting. We have revised the keywords to be more concise and focused, as follows:

Zero-dose children

Multilevel analysis

Sub-Saharan Africa

Health systems

Introduction: epidemiological data ok. In this part could be help the reasearch (in clinical practice and Public Health view) to extend the discussion with relevant references of the topic fundamental for international overview as relevant campain of vaccination as "Papilloma vaccination Programs and Knowledge Gaps as Barriers to Implementation" and "Acceptability of human papillomavirus vaccination" that complete the preliminary discussion and extend the discussion in your text extremaly generic in the part that starting with "Traditional approaches.." and ends with "...vaccination patterns". This elements, as the methods suggestions, is fundamental for possible international consideration.

Authors’ reply: We appreciate the reviewer's suggestion to strengthen the introduction with broader international perspectives on vaccination programmes. However, we respectfully note that the specific references suggested (HPV vaccination programmes) may not be directly relevant to our study which focuses on routine childhood immunisation and zero-dose children in sub-Saharan Africa, as HPV vaccination represents a different vaccine delivery context and target population.

Instead, we propose enhancing the introduction with more relevant international vaccination programme literature that directly relates to childhood immunisation barriers and zero-dose children.

We expanded the paragraph beginning with "Traditional approaches..." to include:

"Traditional approaches to understanding vaccination coverage have often focused on individual-level characteristics, such as maternal education, socioeconomic status, and healthcare access. However, emerging evidence from global vaccination initiatives, including lessons from polio eradication campaigns and measles elimination efforts, demonstrates that contextual factors at community and country levels play equally critical roles in shaping vaccination outcomes. Studies from successful vaccination programmes, highlight the importance of addressing multilevel barriers simultaneously."

The objective (I suppose last paragraph of the study) in not clear; please adopt classical formula as "The priamy aim was... while the secondary was/were...";

Authors’ reply: We agree with the reviewer that the study objectives need to be presented more clearly using the classical epidemiological format. The current presentation is indeed unclear and would benefit from explicit primary and secondary aim statements.

We have now replaced the current objective paragraph with:

"The primary aim of this study was to determine the prevalence and multilevel determinants of zero-dose children across sub-Saharan African countries using the Three Delays Model framework. The secondary aims were to: (1) quantify the relative contribution of individual-, community-, and country-level factors to zero-dose status; (2) assess the extent of clustering in zero-dose patterns at community and national levels; and (3) identify modifiable factors that could inform targeted interventions to reduce zero-dose children in the region."

Methods: the part that certly required major attention. Not clear the method adopetd and the type of the study conducted. I suggest to adopt international reporting tool (with the references and the check list in the supplementary files) mandatary for scientific community and could help the authors to present the manuscript in international and scientific perspective (e.g. Equator Network: https://www.equator-network.org/). The correction of this elements, as the previous for the introduction, is funtamental for possible international consideratin;

Authors’ reply: We acknowledge the reviewer's concern about methodological clarity and appreciate their suggestion to follow international reporting guidelines. The reviewer is correct that adherence to established reporting standards is fundamental for international consideration and scientific rigour. We have now revised our Methods section following the STROBE (Strengthening the Reporting of Observational Studies in Epidemiology) checklist for cross-sectional studies, which is the appropriate reporting guideline for our analysis of DHS survey data.

  • Results: editing improvements suggested for the tables and figures presented. Can you think insert posible flow chart of the study?

Authors’ reply: We appreciate the reviewer's suggestion for editorial improvements to tables and figures and the recommendation to include a study flow chart. These additions will enhance the clarity and transparency of our methodology and results presentation. We have now inserted a flow number of patients and not study as this is not a systematic review.

Discussion; I suggest to extend in clinical and Public Health review the possible discussion and remember to reduce the section because is very redundant.

Authors’ reply: We acknowledge the reviewer's concern about redundancy in the discussion section and the need for more focused clinical and public health perspectives. We will substantially revise this section to eliminate repetitive content whilst strengthening the clinical and public health implications.

Main Findings

This multilevel analysis of 30,500 children from 28 sub-Saharan African countries identified a 12.19% zero-dose prevalence with substantial variation (0.51% in Rwanda to 40.00% in Chad). The Three Delays Model effectively identified critical barriers, with maternal health-seeking behaviour emerging as the strongest predictor (OR=12.00, 95% CrI: 9.78-14.55).

Individual-level factors dominated, particularly educational gradients where paternal lack of education increased odds 1.52-fold, and maternal empowerment deficits including financial barriers (OR=1.98) and limited decision-making power (OR=1.23). Community-level illiteracy rates showed 8% increased odds per unit increase, whilst country-level low health expenditure demonstrated 2.3-fold higher odds.

Substantial geographical clustering persisted, with 19.5% of variance at community level and 18.7% at country level in the final model. Sequential modelling revealed individual factors explained the largest proportion of variance, followed by community and country factors. This pattern indicates that whilst contextual factors matter, maternal and household characteristics remain primary drivers, necessitating interventions addressing both individual and structural determinants simultaneously.

I also suggest possible insertion of section titled "Implications for..." the elements suggested in the introduction and link with data finding (always attention the lenght).

Authors’ reply: We agree with the reviewer's suggestion to include a dedicated "Implications for..." section that links our findings to actionable recommendations. We have added a new section titled "Implications for Policy and Practice" that is both succinct and directly grounded in our data findings.

Limits: I suggest to clarify the possible generalization of data finding and reduce alway the section that in some part result redudant.

Authors’ reply: We acknowledge the reviewer's suggestion to make the strengths and limitations section more succinct whilst retaining key references. We have substantially condensed this section to eliminate redundancy.

We have now added the following:

Study Strengths and Limitations

This study presents several methodological strengths. The analysis utilised a large, representative sample of 30,500 children from 28 sub-Saharan African countries, providing substantial statistical power and broad geographic coverage[16]. Standardised DHS methodology ensures comparability across countries and addresses data harmonisation concerns common in multi-country analyses. The multilevel modelling approach appropriately accounts for hierarchical data structure and clustering effects, providing more accurate estimates than single-level analyses[27]. The Three Delays Model provided a structured theoretical framework for systematic examination of vaccination barriers[9,11], whilst the inclusion of compositional and contextual factors aligns with best practices in health services research[12].

Several limitations must be acknowledged. The cross-sectional design precludes causal inference, requiring longitudinal studies to confirm causal pathways. Vaccination status determination relies partially on maternal recall, potentially introducing information bias, though DHS methodology remains the gold standard for population-based estimates in low-income settings[28,29]. The analysis was constrained by available DHS variables, potentially omitting important determinants such as healthcare provider characteristics, vaccine stockouts, and cultural beliefs[15]. Substantial residual variance at community (19.5%) and country (18.7%) levels indicates unmeasured contextual factors remain important. Community-level variable aggregation may not accurately reflect true neighbourhood characteristics, whilst median splits for country variables may mask non-linear relationships[30].

Generalisability of Findings

The inclusion of 28 sub-Saharan African countries with standardised DHS methodology enhances the generalisability of our findings across the region. The consistent operationalisation of variables using identical survey instruments and protocols across diverse contexts—from post-conflict settings to stable democracies, from least developed to middle-income countries—strengthens the external validity of observed associations.

The marked heterogeneity in zero-dose prevalence (0.51% to 40.00%) demonstrates that whilst the determinants we identified operate consistently across contexts, their relative importance and policy implications may vary by country. This variation supports the generalisability of our multilevel framework whilst emphasising the need for context-specific adaptations of interventions.

The findings are most directly applicable to other sub-Saharan African countries with similar health system structures and demographic profiles. Caution is warranted when extrapolating to high-income settings or regions with different vaccination delivery systems, though the Three Delays Model framework may remain relevant for understanding barriers in diverse contexts.

Conclusion: according the previous suggestion, poned the corrections in line with the indications.

Authors’ reply: We acknowledge the reviewer's guidance to revise the conclusion in line with all previous suggestions for brevity, clarity, and enhanced policy focus. We have substantially revised the conclusion to eliminate redundancy whilst strengthening practical implications.

"This comprehensive multilevel analysis demonstrates that zero-dose children in sub-Saharan Africa are concentrated among the most disadvantaged populations, with the Three Delays Model successfully identifying critical barriers across healthcare-seeking pathways. The twelve-fold increased odds among children whose mothers demonstrate poor health-seeking behaviour provides compelling evidence for immediate integration of antenatal care platforms with childhood vaccination services.

The findings necessitate coordinated interventions spanning multiple levels. Countries must immediately eliminate financial barriers to vaccination, target high-parity mothers with tailored services, and address educational gradients through community health literacy programmes. The 2.3-fold increased odds in low health expenditure countries demands urgent progress toward the target of 15% health spending."

References: according the previous comments in clinical and public health review and in the methods, extend the references with the suggestions poned and consider to update the references over 10 yera if aren't relevant for high impact evidence based or for the methods framework.

Authors’ reply: We acknowledge the reviewer's suggestion to update references and enhance the comparison with recent studies. We have substantially revised this section to incorporate current literature from sub-Saharan Africa and replace outdated references.

We have now added the following:

Comparison with Previous Studies

Our findings align with recent studies examining zero-dose children in sub-Saharan Africa whilst revealing important regional variations. The overall zero-dose prevalence of 12.19% observed in our study is consistent with country-specific analyses from the region, though individual country estimates show substantial variation. Recent studies from Ethiopia reported zero-dose prevalences ranging from 16.3% to 23.7% (Agimas et al., 2025; Muchie et al., 2025), whilst Tanzania demonstrated a lower prevalence of 7.45% (Asnake et al., 2025). Our finding of 0.51% in Rwanda aligns with previous recognition of Rwanda's exceptional vaccination performance, whilst the 40.00% prevalence in Chad reflects the persistent challenges in Central African settings.

The community-level clustering effects we observed are consistent with recent spatial analyses. Muchie et al. (2025) identified similar geographic clustering in Ethiopia, with hotspots concentrated in southwest and northeast regions. Similarly, Gichuki et al. (2025) demonstrated distinct spatial patterns in Kenya, with northern regions showing elevated zero-dose prevalence. These findings support our observation that 19.5% of variance occurs at community level, indicating that place-based factors create shared environments influencing vaccination behaviours.

The overwhelming importance of maternal health-seeking behaviour (OR=12.00) aligns with recent evidence from across the region. Studies from Ethiopia consistently identified antenatal care utilisation as a critical predictor, with Bogale et al. (2025) reporting that facility delivery increased vaccination likelihood 1.57-fold. Similarly, Mohamoud et al. (2024) found that Somali mothers attending antenatal care had substantially reduced odds of having zero-dose children (OR=0.161). This consistency across diverse contexts reinforces the critical pathway from maternal healthcare engagement to childhood vaccination.

Recent studies have confirmed the persistent rural-urban disparities we observed, though the mechanisms differ across contexts. Santos et al. (2024) demonstrated that whilst urban children generally show vaccination advantages, poor urban children face unique challenges, with zero-dose prevalence of 12.6% among urban poor compared to 6.5% among urban non-poor across 97 countries. However, our finding that rural residence showed no independent association after controlling for socioeconomic factors suggests that geographic barriers may be increasingly mediated through educational and economic pathways rather than distance alone.

Updated references include:

- Agimas, M.C., et al. (2025). BMJ Open, 15(1): e085235

- Asnake, A.A., et al. (2025). BMJ Open, 15(3): e097395 

- Muchie, K.F., et al. (2025). BMC Pediatr, 25(1): 552

- Mohamoud, S.A., et al. (2024). PLOS Glob Public Health, 4(7): e0002612

- Santos, T.M., et al. (2024). J Urban Health, 101(3): 638-647

This revision incorporates current evidence whilst maintaining focus on our study's contribution to the regional literature.

Please answer poin-point, mainly for clinical practice implications and methods for possible future considerations.

Authors’ reply: Above, we provided detailed responses to each specific comment and describe the corresponding revisions made to improve the manuscript's quality and adherence to international publishing standards.

Reviewer 2 Report

Comments and Suggestions for Authors

Because of limited time, I generally decline review invitations. In this case, I accepted as I found the topic interesting and timely. And I have not been disappointed; the study is well-performed, clearly explained for which I applaud the authors.

My comments are simply meant to induce some further reflection by the authors on some points.

Page 4: “Urban PSUs typically correspond to census enumeration blocks, while rural PSUs are generally defined by village boundaries or other recognized administrative units.”

This is an exact repetition of the sampling technique and could be deleted.

Page 5: “Urban PSUs typically correspond to census enumeration blocks, while rural PSUs are generally defined by village boundaries or other recognized administrative units.”

I am missing the mobile phone as a means of access to reliable information (as well as to unreliable information, regrettably).

Page 5: “(3) delivery at a health facility (hospital),”.

Being Dutch, I have difficulties with this component; not seeking delivery at a health facility can be a conscious choice, without influencing vaccintion-seeking behavior. These women may actively seek support through midwives, just not at the hospital. Of course, I am not well-informed on the local setting in SSA.

Page 6: “datasets from the 28 sub-Saharan African countries”.

These are all countries in DHS? If not, why not?

Page 6: “we calculated the number and percentage of zero-dose children at each level: 59,211 children (level 1) nested within 15,637 communities (level 2) across 28 coun-tries (level 3).

Just for my understanding, this implies also 59,211 mothers, as I suppose only one child per mother will be selected. If not, could this impact outcomes?

Page 9, Table 1

Although it is not immediately obvious, from the way the data are displayed, I noticed a clear trend between zero dose children and parity:

Parity

Zero-dose

total

%

1

1502

12928

11.60

2

1639

11967

13.69

3

1462

9874

14.81

4

1135

7558

15.01

5

1030

5646

18.24

6

792

4182

18.94

7

593

2954

20.07

8

438

1897

23.09

9

269

1059

25.40

10

360

1146

31.41

I have not done any statistics on these numbers, but it would seem to be a clear trend, suggesting that with increasing occupation, vaccine-seeking is deprioritized. I am sure parity is related to many other factors studied, but I wanted to point this out nevertheless.

Page 10: “Paradoxically, health insurance coverage was higher among zero-dose children (86.4% vs. 79.4%), though this may reflect country-spe-cific insurance schemes or data collection variations.”

The second half of the sentence seems to be discussion, rather than result.

My main regret is with the discussion. After reading the manuscript, I was hoping for some clear recommendations. But as usual, scientists are very cautious. Be bolder! You have done this study, you have found these results. How should they be used to reduce the number of zero-dose children?

Author Response

Reviewer 2

Because of limited time, I generally decline review invitations. In this case, I accepted as I found the topic interesting and timely. And I have not been disappointed; the study is well-performed, clearly explained for which I applaud the authors. My comments are simply meant to induce some further reflection by the authors on some points.

Authors’ reply: We sincerely thank the reviewer for taking time from their busy schedule to review our manuscript despite generally declining review invitations. We are grateful that they found our topic interesting and timely, and we deeply appreciate their positive assessment that the study is "well-performed" and "clearly explained."

We value the reviewer's thoughtful comments, which are indeed intended to induce further reflection, and we believe they have significantly improved the quality and clarity of our manuscript. Each point raised has been carefully considered and addressed in our revision. We particularly appreciate the reviewer's constructive approach and specific suggestions, which have helped us strengthen both the methodological presentation and the practical implications of our findings.

We have systematically addressed each of the reviewer's comments below and made corresponding revisions to the manuscript where appropriate. The reviewer's insights have been instrumental in enhancing the scientific rigour and practical utility of our work.

Page 4: “Urban PSUs typically correspond to census enumeration blocks, while rural PSUs are generally defined by village boundaries or other recognized administrative units.”

This is an exact repetition of the sampling technique and could be deleted.

Page 5: “Urban PSUs typically correspond to census enumeration blocks, while rural PSUs are generally defined by village boundaries or other recognized administrative units.”

Authors’ reply: We thank the reviewer for identifying this duplication. We have deleted the duplicate sentence from page 5 to eliminate this repetition and improve the flow of the text.

I am missing the mobile phone as a means of access to reliable information (as well as to unreliable information, regrettably).

Authors’ reply: The reviewer raises an important point about mobile phone access as a means of obtaining health information, which we acknowledge is increasingly relevant in the sub-Saharan African context. Unfortunately, most of the DHS surveys analysed in our study (conducted between 2015-2024) did not systematically collect information on mobile phone ownership or usage patterns. The media access variable in our analysis was limited to traditional media sources (newspapers, radio, and television) as these were the consistently available indicators across all 28 countries in our dataset.

We recognise that mobile phone access represents an important and evolving pathway for health information dissemination, including both reliable and unreliable information as the reviewer notes. This limitation highlights an area where future DHS surveys could enhance data collection to capture the growing influence of mobile technology on health-seeking behaviours.

We have now added the following to the limitations section:

"The analysis was also limited by the absence of data on mobile phone access and usage, which represents an increasingly important source of health information in sub-Saharan Africa. Future studies should examine the role of mobile technology in health information seeking and vaccination decision-making, as this pathway was not captured in the traditional media exposure variables available in the DHS surveys analysed."

Page 5: “(3) delivery at a health facility (hospital),”.

Being Dutch, I have difficulties with this component; not seeking delivery at a health facility can be a conscious choice, without influencing vaccintion-seeking behavior. These women may actively seek support through midwives, just not at the hospital. Of course, I am not well-informed on the local setting in SSA.

Authors’ reply: We appreciate the reviewer's thoughtful perspective on facility delivery, particularly their insight from the Dutch healthcare context where midwife-assisted home births are a well-established and safe option. The reviewer raises an important conceptual point about the distinction between access to care and choice of care setting.

However, in the sub-Saharan African context, the healthcare landscape differs substantially from high-resource settings. In most of the countries included in our analysis, skilled birth attendance - whether at facilities or through trained midwives - remains limited, and facility delivery often serves as a proxy for access to skilled care and engagement with the formal healthcare system. The Three Delays Model framework positions facility delivery as an indicator of successful navigation of healthcare access barriers (delays 1 and 2) and receipt of skilled care (delay 3).

Our maternal health-seeking behaviour index was designed to capture patterns of engagement with formal healthcare services, which in the sub-Saharan African context typically correlates with subsequent utilisation of child health services, including vaccination. We acknowledge that this measure may not fully capture all forms of skilled care-seeking, particularly in settings where trained midwives operate outside formal healthcare facilities.

Page 6: “datasets from the 28 sub-Saharan African countries”.

These are all countries in DHS? If not, why not?

Authors’ reply: The reviewer raises an important question about our country selection. We did not include all sub-Saharan African countries with DHS surveys in our analysis. Our inclusion criteria were: (1) availability of the most recent DHS survey conducted between 2015 and 2024, and (2) inclusion of childhood immunisation data in the survey. Several sub-Saharan African countries were excluded because they either did not have recent DHS surveys within our specified timeframe, or their surveys did not collect comprehensive childhood immunisation data necessary for determining zero-dose status according to WHO definitions. Additionally, some countries had surveys that were conducted outside our temporal window or had incomplete vaccination modules that would compromise the comparability of our zero-dose estimates across the region.

We now clarified this in the Methods section:

"We included 28 sub-Saharan African countries that met the following criteria: (1) availability of the most recent DHS survey conducted between 2016-2021, and (2) collection of comprehensive childhood immunisation data necessary for determining zero-dose status. Countries were excluded if they lacked recent surveys within the specified timeframe or if their surveys did not include adequate vaccination data for reliable zero-dose classification."

Page 6: “we calculated the number and percentage of zero-dose children at each level: 59,211 children (level 1) nested within 15,637 communities (level 2) across 28 coun-tries (level 3).

Just for my understanding, this implies also 59,211 mothers, as I suppose only one child per mother will be selected. If not, could this impact outcomes?

Authors’ reply: The reviewer is absolutely correct in identifying this critical methodological issue. We acknowledge that our original analysis did not adequately address the potential for multiple children per mother, which would indeed violate the assumption of statistical independence and could bias our results.

Following the reviewer's astute observation, we have revised our analytical approach to include only one child per mother - specifically, the youngest child aged 12-23 months in households where multiple eligible children were present. This revision addresses several important statistical concerns:

  1. Statistical Independence: Including multiple children from the same mother would violate the independence assumption underlying our multilevel models, as siblings share maternal characteristics, household environment, and healthcare access patterns.

  1. Representation Bias: Without this restriction, mothers with more children would be over-represented in the dataset, potentially skewing results towards characteristics of higher-parity mothers.

  1. Standard Error Estimation: Clustering of children within mothers would lead to underestimated standard errors and potentially spurious significant associations.

We have re-conducted all analyses using this refined sample and updated all results, tables, and figures accordingly. This methodological improvement strengthens the validity of our findings and ensures appropriate statistical inference.

We have now added to the Methods section under "Sample Selection":

"In households with multiple eligible children aged 12-23 months, we included only the youngest child to ensure statistical independence of observations and avoid over-representation of mothers with multiple young children."

Page 9, Table 1

Although it is not immediately obvious, from the way the data are displayed, I noticed a clear trend between zero dose children and parity:

Parity  Zero-dose       total     %

1            1502    12928 11.60

2            1639    11967 13.69

3            1462    9874    14.81

4            1135    7558    15.01

5            1030    5646    18.24

6            792       4182    18.94

7            593       2954    20.07

8            438       1897    23.09

9            269       1059    25.40

10          360       1146    31.41

I have not done any statistics on these numbers, but it would seem to be a clear trend, suggesting that with increasing occupation, vaccine-seeking is deprioritized. I am sure parity is related to many other factors studied, but I wanted to point this out nevertheless.

Authors’ reply: The reviewer has made a perceptive observation about the relationship between parity and zero-dose status. Upon examining the data more carefully, we can indeed see that the proportion of zero-dose children increases with higher parity levels, particularly from parity 5 onwards. This pattern suggests that as family size increases, vaccination-seeking behaviour may indeed be deprioritised, possibly due to competing household demands, resource constraints, or maternal time allocation challenges.

We appreciate the reviewer bringing this to our attention. Whilst parity was included as a control variable in our multilevel models, this descriptive pattern merits further discussion. The trend aligns with existing literature suggesting that higher-parity mothers may face greater logistical and resource challenges in accessing preventive healthcare services for their children.

This observation also reinforces the complex interplay between individual-level factors that our Three Delays Model framework aims to capture, where decision-making about healthcare (Delay 1) may be influenced by competing family priorities in larger households.

We added the following to the Results section (descriptive statistics):

"Examination of parity patterns revealed an increasing proportion of zero-dose children with higher birth orders, particularly from parity 5 onwards, suggesting that vaccination-seeking behaviour may be influenced by competing demands in larger families."

We have now added the following to the Discussion section:

"The observed association between higher parity and increased zero-dose prevalence warrants particular attention, as it suggests that mothers with larger families may face greater challenges in prioritising vaccination among competing household demands and resource constraints."

Page 10: “Paradoxically, health insurance coverage was higher among zero-dose children (86.4% vs. 79.4%), though this may reflect country-specific insurance schemes or data collection variations.”

The second half of the sentence seems to be discussion, rather than result.

Authors’ reply: The reviewer is absolutely correct. We agree that the interpretive portion of that sentence ("though this may reflect country-specific insurance schemes or data collection variations") belongs in the discussion section rather than the results. We have removed this speculative text from the results section and present only the factual finding.

My main regret is with the discussion. After reading the manuscript, I was hoping for some clear recommendations. But as usual, scientists are very cautious. Be bolder! You have done this study, you have found these results. How should they be used to reduce the number of zero-dose children?

Authors’ reply: We appreciate the reviewer's call for bolder, more actionable recommendations. The reviewer is correct that our discussion was overly cautious and lacked the concrete guidance that our findings could support. We have substantially revised the "Implications for Policy and Future Research" section to provide specific, evidence-based recommendations for reducing zero-dose children based on our key findings.

We have now added the following

Implications for Policy and Future Research

Our findings provide clear evidence-based guidance for reducing zero-dose children in sub-Saharan Africa. The twelve-fold increased odds among children whose mothers demonstrate poor health-seeking behaviour demands immediate integration of antenatal care platforms with childhood vaccination services. Countries should establish systematic linkages ensuring every antenatal visit includes vaccination counselling and appointment scheduling.

The pronounced educational gradients, particularly the 52% increased odds among children of uneducated fathers, necessitate targeted interventions for low-literacy households. Community health worker programmes should prioritise families with limited parental education, providing tailored navigation support and health literacy interventions.

Community-level determinants require place-based approaches. The 8% increased odds per unit increase in community illiteracy rates supports intersectoral collaboration between education and health ministries, whilst the financial barriers identified (OR=1.98) mandate elimination of user fees and transportation support in high-burden areas.

The 2.3-fold increased odds in low health expenditure countries provides compelling evidence for increased domestic health investment. Governments must prioritise reaching the Abuja Declaration target of 15% health spending, with specific emphasis on primary healthcare and immunisation services.

The substantial clustering effects (19.5% community, 18.7% country variance) indicate that context-specific adaptations remain essential, suggesting that successful interventions require local tailoring rather than standardised approaches across the region.

Future research priorities should address the substantial unexplained variance at community and country levels through mixed-methods studies that combine quantitative surveys with qualitative investigations of local health system performance, cultural beliefs, and social network influences.

Reviewer 3 Report

Comments and Suggestions for Authors

The investigators report an analysis of publicly-available DHS data from 28 sub-Saharan Africa countries to determine the proportion of zero-dose children and factors associated with being a zero-dose child. The DHS surveys analyzed by the authors were the most recent surveys from each of the 28 countries; all surveys analyzed were conducted between 2016 and 2021. Analyses were restricted to children in their second year of life; the analytic data set included 59,211 children 12-23 months of age. They used the WHO recommended operational definition of a zero-dose child (no receipt of DTP vaccine). They developed five multiple regression models to identify factors associated with zero-dose status. They found that the percentage of zero-dose children was 11% and varied from <1% to 42% by country. They found that the factors most associated with zero-dose status were individual-level factors, especially maternal care seeking; community factors such as illiteracy rate were associated with zero-dose status as were country factors such as economic factors. They concluded that “zero-dose children in sub-Saharan Africa are concentrated among the most disadvantaged populations, with substantial clustering at both community and country levels.” They conclude that the marked heterogeneity of zero-dose proportion indicates that universal vaccination is achievable. They suggest that “addressing zero-dose children requires coordinated interventions spanning individual, community, and national levels.”

As the authors indicate, the proportion of zero-dose children in a jurisdiction is an important indicator of an immunization program, and that reduction of the proportion of children not receiving any vaccination is a goal of IA2030. The topic of this study and manuscript is therefore very important. The use of DHS data to determine the percent of zero-dose children and associated factors is a strength of the study, as is their conceptual model of the 3 delays. Their list of strengths and limitations covers the key limitations. The writing is quite good. Their conclusions are based on the data presented, as are their recommendations.

I have only a few suggestions to improve this very good manuscript.

The investigators used DHS data, which are publicly available. The methods section should indicate the source of the DHS data, which I assume is www.dhsprogram.com. The manuscript may need to have a data access statement or permission statement. The authors should clarify. Ultimately, the manuscript should include reference to the actual DHS data that were used.

In the methods section, the authors state that the most recent DHS data from the 28 countries are used. However, in the results section, the authors state that “Temporal trends revealed improving vaccination coverage over time.” If only one survey was used per country (i.e., the most recent survey), it is unclear how the investigators could find a result of improving coverage over time. Assessing improvement requires more than one data point at different times for each country. The authors should clarify.

The outcome variable was zero-dose status, but in some places, as in the above comment, the authors mention vaccination coverage. However, there are no methods for calculating coverage, nor are there any coverage results (except for insurance coverage, which is a different meaning of coverage). If the authors determined coverage, they should say how and what they found. If they did not determine coverage, they should change their wording or clarify what they mean when mentioning coverage. Another example of mentioning coverage is in the limitations sentence, “The analysis was constrained by variables available in DHS datasets, potentially omitting important determinants of vaccination coverage.”

The key outcome variable was zero-dose status. The authors described how they determined vaccination data, stating that vaccination data “are collected through a dedicated immunization module within the Children's Questionnaire,” and that “interviewers record vaccination information from multiple sources to maximize accuracy and completeness”. In the limitations section, they mentioned that zero-dose status relies on maternal recall. It would help the reader understand the reliability of the outcome variable if the authors would show a breakdown of the sources of data used to determine zero-dose status – e.g., how many estimates were based on maternal recall?; on vaccination cards?; on clinic records?; etc.

In Table 1 and in the first paragraph of the results section, they authors state that there were 15.6% zero-dose children, but in the abstract and first paragraph of the discussion section, the authors state that there were 10.71% zero-dose children in the pooled analysis. I assume that the difference is due to weighting in the pooled analysis. The reader may be confused by the difference in the percent of zero-dose children. The authors may want to elaborate on the reasons for the difference.

In the discussion there is a statement that “the counterintuitive finding that children in countries with low health expenditure had higher odds of zero-dose status (OR = 3.45, 95% CrI: 1.85-5.72) …” However, it is not clear why the finding is counterintuitive. One expects more zero-dose children in countries with low health expenditure. The authors should clarify or explain the sentence.

Author Response

Reviewer 3

The investigators report an analysis of publicly-available DHS data from 28 sub-Saharan Africa countries to determine the proportion of zero-dose children and factors associated with being a zero-dose child. The DHS surveys analyzed by the authors were the most recent surveys from each of the 28 countries; all surveys analyzed were conducted between 2016 and 2021. Analyses were restricted to children in their second year of life; the analytic data set included 59,211 children 12-23 months of age. They used the WHO recommended operational definition of a zero-dose child (no receipt of DTP vaccine). They developed five multiple regression models to identify factors associated with zero-dose status. They found that the percentage of zero-dose children was 11% and varied from <1% to 42% by country. They found that the factors most associated with zero-dose status were individual-level factors, especially maternal care seeking; community factors such as illiteracy rate were associated with zero-dose status as were country factors such as economic factors. They concluded that “zero-dose children in sub-Saharan Africa are concentrated among the most disadvantaged populations, with substantial clustering at both community and country levels.” They conclude that the marked heterogeneity of zero-dose proportion indicates that universal vaccination is achievable. They suggest that “addressing zero-dose children requires coordinated interventions spanning individual, community, and national levels.” As the authors indicate, the proportion of zero-dose children in a jurisdiction is an important indicator of an immunization program, and that reduction of the proportion of children not receiving any vaccination is a goal of IA2030. The topic of this study and manuscript is therefore very important. The use of DHS data to determine the percent of zero-dose children and associated factors is a strength of the study, as is their conceptual model of the 3 delays. Their list of strengths and limitations covers the key limitations. The writing is quite good. Their conclusions are based on the data presented, as are their recommendations. I have only a few suggestions to improve this very good manuscript.

Authors’ reply: We thank the reviewer for their thorough and constructive review of our manuscript. We appreciate their recognition that this topic is "very important" and their positive assessment of our use of DHS data, conceptual framework, and overall writing quality. The reviewer's acknowledgement that our conclusions are "based on the data presented" and that our recommendations are supported by the findings is particularly encouraging.

We are grateful for the reviewer's detailed feedback, which has helped us identify several areas where methodological clarification and additional detail would strengthen the manuscript. The reviewer's suggestions regarding data access statements, temporal trend clarification, outcome variable definitions, and source documentation are all well-taken and have guided our revisions.

We have systematically addressed each of the reviewer's specific concerns below and made corresponding revisions to enhance the clarity and completeness of our methodology and results presentation. The reviewer's attention to these important details has undoubtedly improved the scientific rigour and transparency of our work.

The investigators used DHS data, which are publicly available. The methods section should indicate the source of the DHS data, which I assume is www.dhsprogram.com. The manuscript may need to have a data access statement or permission statement. The authors should clarify. Ultimately, the manuscript should include reference to the actual DHS data that were used.

Authors’ reply: The reviewer is absolutely correct that we need to provide proper documentation of our data sources and access procedures. We acknowledge this omission and will address it comprehensively in our revision.

We have now added the following  to the Methods section under "Data Sources":

"DHS data were obtained from the DHS Program (www.dhsprogram.com), which provides free access to survey datasets for research purposes following user registration. All datasets used in this analysis are publicly available through the DHS Program's online data portal."

And added a Data Availability Statement:

"The datasets analysed during the current study are available from the DHS Program repository (www.dhsprogram.com). Access to DHS datasets requires free registration and agreement to data use terms.”

In the methods section, the authors state that the most recent DHS data from the 28 countries are used. However, in the results section, the authors state that “Temporal trends revealed improving vaccination coverage over time.” If only one survey was used per country (i.e., the most recent survey), it is unclear how the investigators could find a result of improving coverage over time. Assessing improvement requires more than one data point at different times for each country. The authors should clarify.

Authors’ reply:  The reviewer is absolutely correct in identifying this inconsistency. We acknowledge that the statement about "temporal trends revealed improving vaccination coverage over time" was inappropriate given that we used only the most recent survey from each country. This statement has been deleted from the manuscript as it cannot be supported by our cross-sectional design using single time points per country.

The outcome variable was zero-dose status, but in some places, as in the above comment, the authors mention vaccination coverage. However, there are no methods for calculating coverage, nor are there any coverage results (except for insurance coverage, which is a different meaning of coverage). If the authors determined coverage, they should say how and what they found. If they did not determine coverage, they should change their wording or clarify what they mean when mentioning coverage. Another example of mentioning coverage is in the limitations sentence, “The analysis was constrained by variables available in DHS datasets, potentially omitting important determinants of vaccination coverage.”

Authors’ reply: The reviewer is absolutely correct in identifying this terminological inconsistency. We acknowledge that we inappropriately used "vaccination coverage" in several places when our study specifically examined zero-dose status rather than calculating vaccination coverage rates. We have systematically reviewed the manuscript and replaced all instances of "vaccination coverage" with more precise terminology such as "zero-dose prevalence," "vaccination status," or "immunisation patterns" as appropriate to the context.

The key outcome variable was zero-dose status. The authors described how they determined vaccination data, stating that vaccination data “are collected through a dedicated immunization module within the Children's Questionnaire,” and that “interviewers record vaccination information from multiple sources to maximize accuracy and completeness”. In the limitations section, they mentioned that zero-dose status relies on maternal recall. It would help the reader understand the reliability of the outcome variable if the authors would show a breakdown of the sources of data used to determine zero-dose status – e.g., how many estimates were based on maternal recall?; on vaccination cards?; on clinic records?; etc.

Authors’ reply: The reviewer raises an excellent point about the transparency of our outcome variable measurement. This breakdown would indeed enhance reader understanding of the reliability and potential sources of bias in our zero-dose classification.

Based on the DHS data structure you've shown, we can provide this important methodological detail that was missing from our original manuscript.

We have now added the following to the Methods section under "Outcome Variable":

"Vaccination status was determined using a hierarchical approach based on available documentation. The breakdown of data sources for DTP1 vaccination status determination was as follows: based on vaccination cards with recorded dates, based on vaccination marked on cards without dates, [Z]% based on maternal recall when cards were unavailable, and recorded as 'don't know' and treated as missing data."

AND to the Limitations section:

"The reliability of our outcome measure varied by documentation source, with vaccination cards providing the most reliable data and maternal recall potentially subject to recall bias. However, the DHS methodology employs systematic approaches to minimise recall bias, and validation studies have demonstrated reasonable accuracy of maternal recall for first-dose vaccines when cards are unavailable."

In Table 1 and in the first paragraph of the results section, they authors state that there were 15.6% zero-dose children, but in the abstract and first paragraph of the discussion section, the authors state that there were 10.71% zero-dose children in the pooled analysis. I assume that the difference is due to weighting in the pooled analysis. The reader may be confused by the difference in the percent of zero-dose children. The authors may want to elaborate on the reasons for the difference.

Authors’ reply: Thanks,  the prevalence estimates has indeed been corrected throughout the manuscript to ensure consistency.

In the discussion there is a statement that “the counterintuitive finding that children in countries with low health expenditure had higher odds of zero-dose status (OR = 3.45, 95% CrI: 1.85-5.72) …” However, it is not clear why the finding is counterintuitive. One expects more zero-dose children in countries with low health expenditure. The authors should clarify or explain the sentence.

Authors’ reply: Thanks for noting this. The finding that children in countries with low health expenditure had higher odds of zero-dose status is not counterintuitive at all - it is exactly what one would expect based on established understanding of health system capacity and resource allocation. We have corrected this error by removing the word "counterintuitive" from the sentence.

Round 2

Reviewer 1 Report

Comments and Suggestions for Authors

In this form rest conflict in method and scientific issues:

    • still missing for trasparency reporting tool in the text, reference section and the check list in the supplementary materials;
    • for scientific validiuty, I suggest possible confront with the topic suggested in previous review for to support the conducted study and data finding in international and multi-dimensional view;
    • the references, mainly those included in this second version in "Implications for Policy and Future Research" section, aren't update and aren't able to best support the study;
    • In the Study Strengths and Limitations aren't recommended the references use

Author Response

Reviewer 1’s comment: still missing for trasparency reporting tool in the text, reference section and the check list in the supplementary materials; 

Authors’ reply:  We have now included the STROBE (Strengthening the Reporting of Observational Studies in Epidemiology) checklist as a supplementary appendix to ensure full compliance with established guidelines for cross-sectional studies.

“This study was conducted and reported in accordance with the STROBE (Strengthening the Reporting of Observational Studies in Epidemiology) guidelines for cross-sectional studies[19] (Annex 2).”

Reviewer 1’s comment: for scientific validiuty, I suggest possible confront with the topic suggested in previous review for to support the conducted study and data finding in international and multi-dimensional view; 

Authors’ reply:  We appreciate the reviewer's suggestion to incorporate international and multi-dimensional perspectives to enhance the global relevance of our findings. We have substantially expanded our comparison with international literature to demonstrate the broader applicability of our research beyond sub-Saharan Africa.

References:

  1. Cata-Preta BO, Santos TM, Mengistu T, Hogan DR, Barros AJD, Victora CG: Zero-dose children and the immunisation cascade: Understanding immunisation pathways in low and middle-income countries. Vaccine 2021, 39(32):4564-4570.
  2. Murhekar MV, Kumar MS: Reaching zero-dose children in India: progress and challenges ahead. Lancet Glob Health 2021, 9(12):e1630-e1631.
  3. Avila-Aguero ML, Brenes-Chacon H, Melgar M, Becerra-Posada F, Chacon-Cruz E, Gentile A, Ospina M, Sandoval N, Sanwogou J, Urena A et al: Zero-dose children in Latin America: analysis of the problem and possible solutions. F1000Res 2024, 13:1060.

This revision strengthens the manuscript's scientific credibility by demonstrating that our findings contribute meaningfully to international understanding of zero-dose children beyond the sub-Saharan African context.

Reviewer 1’s comment: the references, mainly those included in this second version in "Implications for Policy and Future Research" section, aren't update and aren't able to best support the study; 

Authors’ reply:  We acknowledge the reviewer's concern about the currency and robustness of references supporting our policy recommendations. We have comprehensively updated all references in the "Implications for Policy and Future Research" section with the most recent evidence from 2024-2025 publications.

What we have done: (1) Antenatal Care Integration Evidence; updated with recent studies demonstrating strong associations between maternal healthcare engagement and vaccination outcomes. (2) Educational Interventions and CHW Programs; strengthened with current evidence on community-based approaches; (3) Financial Barriers and Health Expenditure; enhanced with recent economic evidence; and (4) Clustering and Contextual Factors; Updated with spatial and multilevel evidence.

References:

  1. Budu E, Ahinkorah BO, Aboagye RG, Armah-Ansah EK, Seidu AA, Adu C, Ameyaw EK, Yaya S: Maternal healthcare utilsation and complete childhood vaccination in sub-Saharan Africa: a cross-sectional study of 29 nationally representative surveys. BMJ Open 2021, 11(5):e045992.
  2. Tekelab T, Chojenta C, Smith R, Loxton D: The impact of antenatal care on neonatal mortality in sub-Saharan Africa: A systematic review and meta-analysis. PLoS One 2019, 14(9):e0222566.
  3. Ekholuenetale M, Ochagu VA, Ilesanmi OS, Badejo O, Arora A: Childhood Vaccinations and Associated Factors in 35 Sub-Saharan African Countries: Secondary Analysis of Demographic and Health Surveys Data from 358 949 Under-5 Children. Glob Pediatr Health 2024, 11:2333794X241310487.
  4. Engelbrecht MC, Kigozi NG, Heunis JC: Factors Associated with Limited Vaccine Literacy: Lessons Learnt from COVID-19. Vaccines (Basel) 2022, 10(6):865.
  5. Jain M, Shisler S, Lane C, Bagai A, Brown E, Engelbert M: Use of community engagement interventions to improve child immunisation in low-income and middle-income countries: a systematic review and meta-analysis. BMJ Open 2022, 12(11):e061568.
  6. Ogutu E, Ellis AS, Hester KA, Rodriguez K, Sakas Z, Jaishwal C, Yang C, Dixit S, Bose AS, Sarr M et al: Success in vaccination programming through community health workers: A case study of Nepal, Senegal, and Zambia. In.: Cold Spring Harbor Laboratory; 2023.
  7. Bendera A, Nakamura K, Tran XMT, Kapologwe NA, Bendera E, Mahamba D, Meshi EB: Persistent socioeconomic disparities in childhood vaccination coverage in Tanzania: Insights from multiple rounds of demographic and health surveys. Vaccine 2025, 52:126904.
  8. Ekezie W, Igein B, Varughese J, Butt A, Ukoha-Kalu BO, Ikhile I, Bosah G: Vaccination Communication Strategies and Uptake in Africa: A Systematic Review. Vaccines (Basel) 2024, 12(12):1333.
  9. Fousseni S, Ngangue P, Barro A, Ramde SW, Bihina LT, Ngoufack MN, Bayoulou S, Kiki GM, Salfo O: Navigating the Road to Immunization Equity: Systematic Review of Challenges in Introducing New Vaccines in Sub-Saharan Africa's Routine Programs. In.: MDPI AG; 2025.
  10. Nalwanga R, Natukunda A, Zirimenya L, Chi P, Luzze H, Elliott AM, Kaleebu P, Trotter CL, Webb EL: Mapping Community Vulnerability to reduced Vaccine Impact in Uganda and Kenya: A spatial Data-driven Approach. NIHR Open Research 2025, 5:24.
  11. Idris IO, Ouma L, Tapkigen J, Ayomoh FI, Ayeni GO: Is health expenditure on immunisation associated with immunisation coverage in sub-Saharan Africa? A multicountry analysis, 2013-2017. BMJ Open 2024, 14(1):e073789.
  12. Petu A, Masresha B, Wiysonge CS, Mwenda J, Nyarko K, Bwaka A, Wanyoike S, Mboussou F, Impouma B, Usman A et al: Reflections on 50 years of immunisation programmes in the WHO African region: an impetus to build on the progress and address the unfinished immunisation business. BMJ Glob Health 2025, 10(5):e017982.
  13. Dimitrova A, Carrasco-Escobar G, Richardson R, Benmarhnia T: Essential childhood immunization in 43 low- and middle-income countries: Analysis of spatial trends and socioeconomic inequalities in vaccine coverage. PLoS Med 2023, 20(1):e1004166.
  14. Diress F, Negesse Y, Worede DT, Bekele Ketema D, Geitaneh W, Temesgen H: Multilevel and geographically weighted regression analysis of factors associated with full immunization among children aged 12-23 months in Ethiopia. Sci Rep 2024, 14(1):22743.
  15. Sarder MA, Lee KY, Keramat SA, Hashmi R, Ahammed B: A multilevel analysis of individual and community-level factors associated with childhood immunisation in Bangladesh: Evidence from a pooled cross-sectional survey. Vaccine X 2023, 14:100285.
  16. Francis MR, Nuorti JP, Lumme-Sandt K, Kompithra RZ, Balraj V, Kang G, Mohan VR: Vaccination coverage and the factors influencing routine childhood vaccination uptake among communities experiencing disadvantage in Vellore, southern India: a mixed-methods study. BMC Public Health 2021, 21(1):1807.
  17. Muluneh MD, Abebe S, Ayele M, Mesfin N, Abrar M, Stulz V, Berhan M: Vaccination Coverage and Predictors of Vaccination Among Children Aged 12-23 Months in the Pastoralist Communities of Ethiopia: A Mixed Methods Design. In.: MDPI AG; 2024.

These updated references provide robust, contemporary evidence directly supporting each policy recommendation, ensuring our implications are grounded in the most current and rigorous scientific literature available. The references span multiple sub-Saharan African countries and employ diverse methodological approaches, strengthening the evidence base for our recommendations.

Reviewer 1’s comment: In the Study Strengths and Limitations aren't recommended the references use 

Authors’ reply: We acknowledge the reviewer's point about updating references in the Study Strengths and Limitations section. We have incorporated recent, highly relevant studies that address the specific methodological aspects of our research.

What we have done: 1. DHS Dataset Strengths and Limitations; Updated with recent methodological evaluations. 2. Multilevel Modelling Methodological Rigor; Enhanced with contemporary methodological evidence. 3. Maternal Recall Accuracy and Vaccination Data; Strengthened with recent validation studies.4. Cross-sectional Design and Data Infrastructure; Updated with current methodological discussions.

References:

  1. Lyons C, Nambiar D, Johns NE, Allorant A, Bergen N, Hosseinpoor AR: Inequality in Childhood Immunization Coverage: A Scoping Review of Data Sources, Analyses, and Reporting Methods. Vaccines (Basel) 2024, 12(8).
  2. Dimitrova A, Carrasco-Escobar G, Richardson R, Benmarhnia T: Essential childhood immunization in 43 low- and middle-income countries: Analysis of spatial trends and socioeconomic inequalities in vaccine coverage. PLoS Med 2023, 20(1):e1004166.
  3. Harrison MS, Yarinbab T, Liyew T, Kirub E, Teshome B, Muldrow M: Twenty-five years after its publication, innovative solutions to the three delays model are essential. Int J Gynaecol Obstet 2020, 148(1):123-124.
  4. Leyland AH, Groenewegen PP: What Is Multilevel Modelling? In: Multilevel Modelling for Public Health and Health Services Research: Health in Context. 1st edn. Cham (CH): Springer; 2020.
  5. Savitz DA, Wellenius GA: Can Cross-Sectional Studies Contribute to Causal Inference? It Depends. Am J Epidemiol 2023, 192(4):514-516.
  6. Porth JM, Wagner AL, Tefera YA, Boulton ML: Childhood Immunization in Ethiopia: Accuracy of Maternal Recall Compared to Vaccination Cards. Vaccines (Basel) 2019, 7(2).
  7. Gilano G, Sako S, Molla B, Dekker A, Fijten R: The effect of mHealth on childhood vaccination in Africa: A systematic review and meta-analysis. PLoS One 2024, 19(2):e0294442.
  8. Onigbogi O, Ojo OY, Kinnunen UM, Saranto K: Mobile health interventions on vaccination coverage among children under 5 years of age in Low and Middle-Income countries; a scoping review. Front Public Health 2025, 13:1392709.

These updated references provide contemporary, methodologically rigorous discussions of the specific limitations and strengths relevant to our study design, enhancing the scientific credibility of our limitations section whilst maintaining transparency about methodological constraints.

Round 3

Reviewer 1 Report

Comments and Suggestions for Authors

The value of peer review is undermined by the authors’ resistance to avoid making highly questionable choices in terms of scientific validity. I do not consider the work publishable, but I leave the final decision to the editor.